# Pupal behavior emerges from unstructured muscle activity in response to neuromodulation in *Drosophila*

Amicia D Elliott[1,2], Adama Berndt[1], Matthew Houpert[1], Snehashis Roy[1], Robert L Scott[1], Carson C Chow[3], Hari Shroff[2], Benjamin H White[1]*

[1]National Institute of Mental Health, National Institutes of Health, Bethesda, United States; [2]National Institute of Biomedical Imaging and Bioengineering, National Institutes of Health, Bethesda, United States; [3]National Institute of Diabetes and Digestive and Kidney Diseases, National Institutes of Health, Bethesda, United States

**Abstract** Identifying neural substrates of behavior requires defining actions in terms that map onto brain activity. Brain and muscle activity naturally correlate via the output of motor neurons, but apart from simple movements it has been difficult to define behavior in terms of muscle contractions. By mapping the musculature of the pupal fruit fly and comprehensively imaging muscle activation at single-cell resolution, we here describe a multiphasic behavioral sequence in *Drosophila*. Our characterization identifies a previously undescribed behavioral phase and permits extraction of major movements by a convolutional neural network. We deconstruct movements into a syllabary of co-active muscles and identify specific syllables that are sensitive to neuromodulatory manipulations. We find that muscle activity shows considerable variability, with sequential increases in stereotypy dependent upon neuromodulation. Our work provides a platform for studying whole-animal behavior, quantifying its variability across multiple spatiotemporal scales, and analyzing its neuromodulatory regulation at cellular resolution.

**\*For correspondence:**
benjaminwhite@mail.nih.gov

**Competing interests:** The authors declare that no competing interests exist.

## Introduction

A major goal of neuroscience is explaining how nervous systems generate and organize behavior. This requires describing behavior in terms that can be correlated with neural activity. The dynamics of brain activity can be observed in whole brains at single-cell resolution (*Ahrens et al., 2013*; *Ardiel et al., 2017*; *Cong et al., 2017*; *Lemon et al., 2015*; *Nguyen et al., 2016*; *Pulver et al., 2015*), but behavioral dynamics has not been captured at a similar level of detail (*Datta et al., 2019*). While progress in fine-mapping natural behavior, or 'computational ethology' (*Anderson and Perona, 2014*), has benefited from recent advances in visual tracking (*Johnson et al., 2020*), 3D imaging (*Hong et al., 2015*), machine vision (*Dankert et al., 2009*), machine learning (*Kabra et al., 2013*; *Machado et al., 2015*), and image feature extraction (*Berman et al., 2014*; *Mathis et al., 2018*; *Wiltschko et al., 2015*), the primary focus of these efforts has been kinematic, seeking to define anatomical movements at higher resolution. While movements are the essential components of behavior, they are complex products of motor neuron activity, which must balance the contractions of agonist and antagonist muscles and must also promote anatomical rigidity that supports movement. Here, we bridge the gap between motor neuron activity and movement by describing muscle activity at the single-cell level.

Comprehensively monitoring muscle activity in behaving animals is achievable with genetically encoded Ca++ indicators and has been demonstrated at single-cell resolution in hydra (*Szymanski and Yuste, 2019*), roundworms (*Ardiel et al., 2017*), and larval fruit flies

**eLife digest** How do we find out how the brain works? One way is to use imaging techniques to visualise an animal's brain in action as it performs simple behaviours: as the animal moves, parts of its brain light up under the microscope. For laboratory animals like fruit flies, which have relatively small brains, this lets us observe their brain activity right down to the level of individual brain cells.

The brain directs movements via collective activity of the body's muscles. Our ability to track the activity of individual muscles is, however, more limited than our ability to observe single brain cells: even modern imaging technology still cannot monitor the activity of all the muscle cells in an animal's body as it moves about. Yet this is precisely the information that scientists need to fully understand how the brain generates behaviour.

Fruit flies perform specific behaviours at certain stages of their life cycle. When the fly pupa begins to metamorphose into an adult insect, it performs a fixed sequence of movements involving a set number of muscles, which is called the pupal ecdysis sequence. This initial movement sequence and the rest of metamorphosis both occur within the confines of the pupal case, which is a small, hardened shell surrounding the whole animal. Elliott et al. set out to determine if the fruit fly pupa's ecdysis sequence could be used as a kind of model, to describe a simple behaviour at the level of individual muscles.

Imaging experiments used fly pupae that were genetically engineered to produce an activity-dependent fluorescent protein in their muscle cells. Pupal cases were treated with a chemical to make them transparent, allowing easy observation of their visually 'labelled' muscles. This yielded a near-complete record of muscle activity during metamorphosis.

Initially, individual muscles became active in small groups. The groups then synchronised with each other over the different regions of the pupa's body to form distinct movements, much as syllables join to form words. This synchronisation was key to progression through metamorphosis and was co-ordinated at each step by specialised nerve cells that produce or respond to specific hormones.

These results reveal how the brain might direct muscle activity to produce movement patterns. In the future, Elliott et al. hope to compare data on muscle activity with comprehensive records of brain cell activity, to shed new light on how the brain, muscles, and other factors work together to control behaviour.

(*Heckscher et al., 2012*; *Zarin et al., 2019*). However, the application of muscle $Ca^{++}$ imaging to characterize more complex sequences has been constrained by the challenge of tracking behavior in freely moving animals. This problem is resolved in pupal fruit flies where behavior is restricted to the puparium (*Kim et al., 2006*), which can be clarified for optical access to the confined animal. The pupa maintains a fixed position and orientation during movement, and all behavior is executed within a delimited field of view. Pupal $Ca^{++}$ activity imaging of muscles has been previously demonstrated using GCaMP6s by *Diao et al., 2017*, who showed that the bulk $Ca^{++}$ signal collected over ventral muscles exhibits temporal patterns that conform well to known body wall movements.

The behavioral hallmark of pupal development is the *Drosophila* pupal ecdysis sequence, which is one of a general class of behavioral sequences used by insects to molt (*Truman, 2005*; *Zitnan and Adams, 2012*). These sequences typically divide into three principal phases during which the animal first loosens and then sheds its old exoskeleton before expanding its newly secreted one. Ecdysis sequences are strongly dependent on the action of hormones for their initiation and progression, and together with vertebrate reproductive behaviors, they have long served as a useful model of hormone-behavior interactions. Neural control of ecdysis behaviors is typically exercised by a conserved complement of hormones including eclosion hormone, ecdysis triggering hormone (ETH), crustacean cardioactive peptide (CCAP), and Bursicon (*Ewer and Reynolds, 2002*; *White and Ewer, 2014*). The sites of action of these hormones within the nervous system have been principally studied in *Drosophila*, but detailed neuroethological studies have been undertaken in larger insects, such as the locust, *Schistocerca gregaria* (*Hughes, 1980a*; *Hughes, 1980b*; *Hughes, 1980c*), and cricket, *Teleogryllus oceanicus* (*Carlson, 1977*; *Carlson and Bentley, 1977*). Motor program organization of the adult ecdysis sequences in these insects is quite stereotyped, suggesting central control of

motor execution, but sensory feedback can also modulate behavioral performance. While characterization of central nervous system (CNS) activity underlying ecdysis sequences has remained limited in larger insects, progress has been made in *Drosophila* where a fictive ecdysis sequence can be elicited in an excised pupal brain by exposure to ETH (*Diao et al., 2017*; *Kim et al., 2015*; *Kim et al., 2006*; *Mena et al., 2016*). Fictive activity imaged using $Ca^{++}$ indicators grossly correlates with the motor patterns executed during pupal ecdysis, but comprehensively interpreting CNS activity remains a challenge.

The *Drosophila* pupal ecdysis sequence occurs at the onset of metamorphosis (*Figure 1A*) and has been adapted to initiate transformation of the body plan (*Kim et al., 2006*). Although it occurs in the context of the pupal molt and has three principal phases characterized by distinct motor programs, its function in molting is limited to casting off the cuticular linings of the gut and trachea. Its primary function is, instead, to create internal pressure at points along the body to evert adult parts such as the head, legs, and wings (*Denlinger and Zdarek, 1994*). ETH initiates the pupal ecdysis sequence after a long period of behavioral quiescence and targets neurons that express its two receptor isoforms, ETHRA and ETHRB (*Kim et al., 2006*). Neurons expressing ETHRB are largely dispensable during larval life, but are essential for pupal ecdysis (*Diao et al., 2016*). Neurons expressing ETHRA include the neuroendocrine cells that secrete CCAP and Bursicon. These cells are likewise essential at pupal, but not larval, ecdysis (*Clark et al., 2008*; *Park et al., 2003*). They become active approximately 10 min after the onset of pupal ecdysis, and their activation mediates the transition to the next behavioral phase. Understanding how the nervous system transforms such hormonal signals into temporally ordered behavioral sequences will require a more complete description of the motor neuron activity that dictates the behavioral sequences themselves.

Here, we use body-wide fluorescence imaging from the dorsal, lateral, and ventral views to characterize the pupal ecdysis sequence at single-cell resolution. Using improved imaging methods—including a new pan-muscle LexA driver that permits dual imaging of muscle and neuron activity—we identify novel elements of the pupal ecdysis sequence, including previously undescribed movements and a phase of stochastic muscle activity preceding ecdysis. We find that muscle activity exhibits a high degree of variability, with individual muscles recruited stochastically into repeating small ensembles, which we call syllables. Syllable activity is synchronized over anatomical compartments to form movements, which are sufficiently stereotyped to be learned by a convolutional neural network (CNN; for code see https://github.com/BenjaminHWhite/muscle_activity; copy archived at swh:1:rev:66456f6ff61e8faa9fe4b442b91ef3fce3b178f9, *White, 2021*). We can prevent synchronization at specific motor program transitions by suppressing neuromodulatory neurons, which is lethal. The suppression of proprioceptive neurons, which blocks initiation of pupal ecdysis, is also unexpectedly lethal. Overall, our analysis at single-cell resolution reveals a dynamical system in which movements are not rigidly specified but form from variable components subject to neuromodulatory reorganization.

## Results

### Pupal behavior described at cellular resolution

Previous characterization of pupal ecdysis has distinguished three principal phases (*Diao et al., 2017*; *Kim et al., 2006*; *Figure 1A–C*, *Video 1*). The first phase (P1) consists of sustained longitudinal compression ('lifting') of posterior abdominal segments accompanied by unilateral, anteriorly propagating, 'rolling contractions' of the dorsal body wall that alternate left-to-right. The second (P2) features left-to-right alternating lateral 'swinging' movements formed by unilateral, anteriorly directed contractions, while the third (P3) consists of alternating left-right posteriorly directed contractions that change into bilaterally symmetric, backward peristaltic contractions. These phases always proceed in the same order. The movements increase internal pressure to evert the head (i.e., force it out of the body cavity) and push the developing legs and wings to the body surface and elongate them (*Zdarek and Friedman, 1986*). Phalloidin labeling demonstrates that approximately half of larval muscles persist until pupal ecdysis, and all retain innervation by Ib synapses (*Prokop, 2006*; *Figure 1D, E*). The most prominent loss of muscles occurs in the ventral and posterior compartments. Only 5 of 13 larval ventral muscles survive (*Figure 1F vs. G*), and one of these (M12) is absent in posterior segments (*Supplementary file 1*). Also missing from posterior

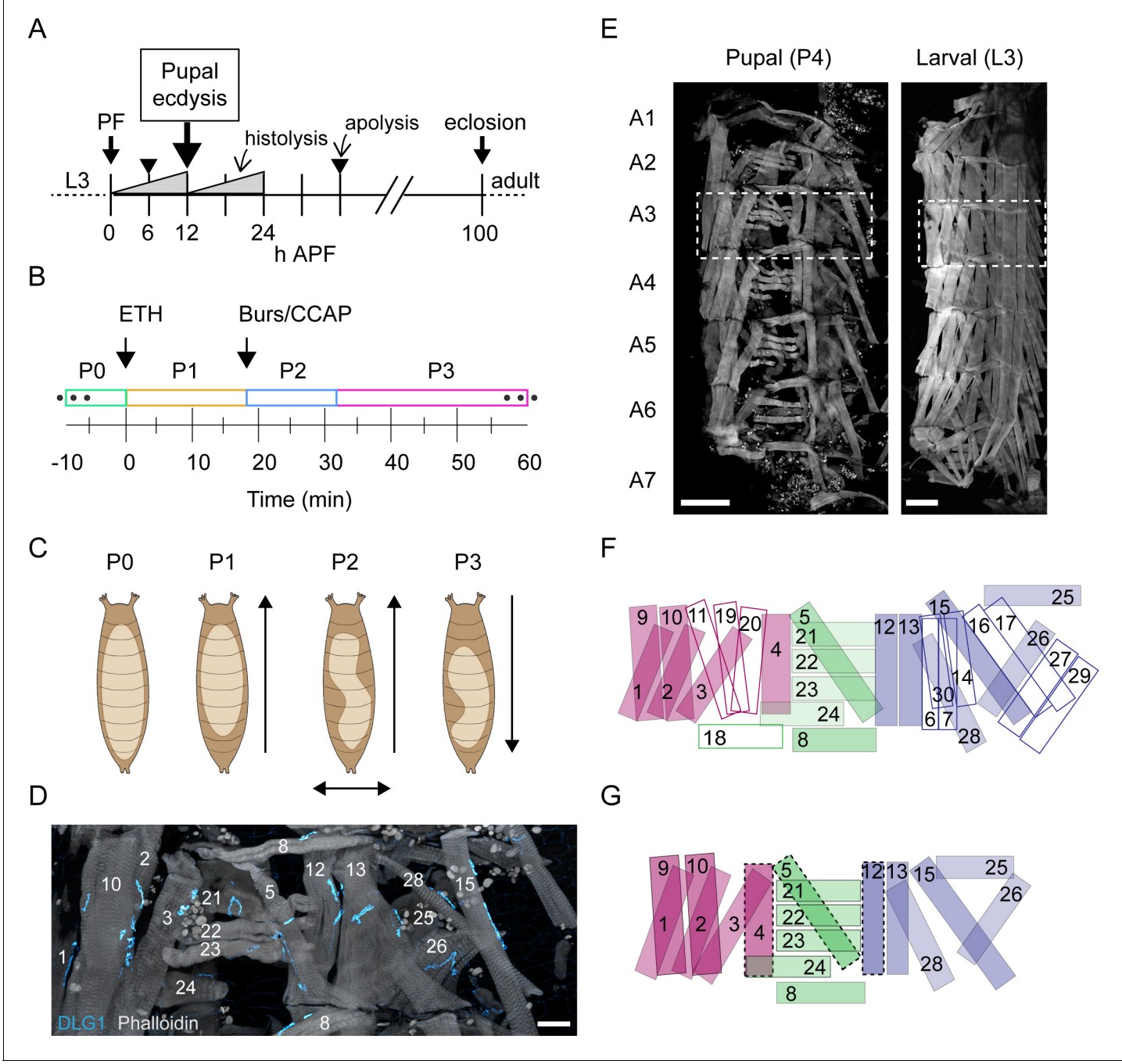

**Figure 1.** Pupal ecdysis behavior and muscle anatomy. (**A**) Timeline of metamorphosis from puparium formation (PF at 0 hr) to eclosion (~100 hr) with the time of pupal ecdysis (arrow) indicated. Other significant events in pupal development include two major periods of tissue histolysis prior to and following pupal ecdysis as well as the times of larval-to-pupal and pupal-to-adult apolysis when separation from the cuticle of the previous developmental stage occurs. L3: third larval instar; APF: after puparium formation. (**B**) Timeline of pupal ecdysis with phases indicated. Arrows indicate times of hormone release. T = 0, onset of P1. (**C**) Schematics show puparium (brown) and pupa (beige) from dorsal view. Arrows indicate directionality of body wall movements for each phase. (**D**) Pupal muscles of hemisegment (HS) A3 stained with phalloidin (gray) and for postsynaptic marker DLG1 (cyan). Scale bar, 25 µm. (**E**) Pupal and larval musculature stained with phalloidin (gray) showing left HSA1–A7 (A3, boxed). Scale bar, 250 µm. (**F**) Larval muscles in a representative HS (e.g., dotted box in (**E**), right; based on six fillets). Outlined muscles degrade before pupal ecdysis. Muscles are labeled and coded blue (ventral), green (lateral), and fuchsia (dorsal). (**G**) Pupal muscles in a representative HS (e.g., dotted box in E, left; based on 17 fillets). Dotted lines indicate those present only in anterior segments. Colors as in (**F**). See also *Video 1*.

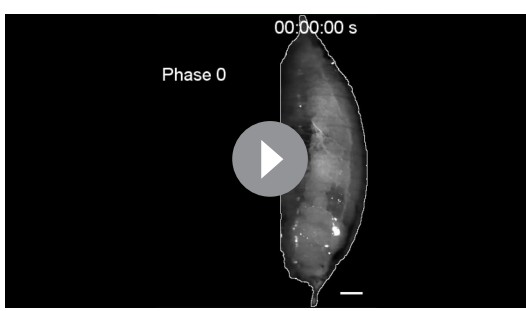

**Video 1.** Pupal ecdysis behavior (lateral view). Bright-field video taken from the lateral side showing a pupa performing the pupal ecdysis sequence. Phases 1–3 are labeled. Sped up to 50 fps. Scale bars, 250 µm. Time, seconds.

https://elifesciences.org/articles/68656#video1

segments are muscles M4 and M5. All five dorsal longitudinal muscles, except M4, are present in all segments, as are five of the six lateral trans-verse muscles. This was consistent across 17 animals, indicating that pupal ecdysis is executed by a standard set of persistent larval muscles.

Although *Drosophila* behavior has been recorded at single-cell resolution, the drivers used to make such recordings express weakly at the pupal stage and cannot be used with Gal4 drivers targeting other cells such as neurons. We identified a striated muscle-specific gene, *l(2) 01289* (aka CG9432), that expresses robustly from early larval stages through adulthood, which we call *hulk* (*hlk*). We generated a Trojan exon insertion in the *hlk* gene that co-expresses LexA: QF and combined it with the $Ca^{++}$ biosensor Lex-A$_{op}$-GCaMP6s to report muscle activity. With this *hlk*-LexA>LexA$_{op}$-GCaMP6s line (i.e., *hlk*>GCaMP6s), we imaged muscle $Ca^{++}$ activity through the clarified puparium for approximately 90 min to capture pupal ecdysis behavior, in vivo (see Materials and methods; *Figure 2*, *Figure 2— figure supplements 1* and *2*).

The muscle activity patterns of animals imaged from the dorsal side match known behaviors, such as the bilateral posterior 'Lifts' and left-right alternating 'rolling contraction' (RollCon) movements of P1 (*Figure 2A*, *Video 2*). Temporal patterns of bulk $Ca^{++}$ activity differentiated phases P1, P2, and P3 (*Figure 2B*). Traces show phase-specific oscillations of varying amplitude and frequency, with individual oscillations conforming to bouts of movement (see Materials and methods). Consistent with *Diao et al., 2017*, the alternating left-right oscillations of P1 persisted through P2 and P3, with coordinated bilateral activity becoming dominant only in P3. Bulk $Ca^{++}$ activity imaged from the lateral side showed the same three phases (*Figure 2C*), and oscillations reflected bouts that included identifiable movements (*Figure 2D*). The Lift is performed by most muscles across the dorsal-ventral (D-V) axis in the posterior segments, while RollCons typically lack activity in the ventral longitudinal muscles 12, 13, and 15 (*Video 3*). Posterior-to-anterior (P-to-A) waves of activity in P1 reverse direction after head eversion in P2 (*Figure 2E*), as previously shown (*Kim et al., 2006*). These data confirm and refine previous observations and demonstrate that the *hlk*>GCaMP6s line accurately reports the ecdysis sequence behaviors.

## Muscle imaging reveals an initial phase of random neurogenic activity

$Ca^{++}$ imaging revealed a phase of random muscle activity prior to the onset of pupal ecdysis (*Figure 2B, C*), which we call Phase 0 (P0). It begins approximately 3 hr before P1 and divides into distinct bouts of muscle activation (*Figure 3A*, *Video 4*). Individual muscle length changes during P0 are small (≤25%) compared to P1–P3 (25–40%; *Figure 3—figure supplement 1A*) and coincide with small body wall twitches rather than coherent movements. Such twitches are also observed during embryonic motor development and are initially myogenic, but later become neurogenic (*Crisp et al., 2008*; *Crisp et al., 2011*). To determine if random P0 muscle activation is myogenic or neurogenic, we created a dual-reporter fly line with *hlk*-LexA driving expression of the red fluorescent $Ca^{++}$ biosensor jRGECO in the muscle and VGlut-Gal4 driving a Synaptotagmin-GCaMP6s fusion protein (Syt-GCaMP6s) in motor neurons, where it localizes to the neuromuscular junction (NMJ, *Figure 3B*). In vivo imaging revealed synaptic $Ca^{++}$ activity at the NMJ 30–40 min prior to the first muscle $Ca^{++}$ response (*Figure 3C*). As P0 progresses, coincident synaptic and muscle activity increases until the onset of P1, when nearly all muscles and their synaptic inputs are synchronously active (*Figure 3D*) except M12, which remains unresponsive to input until P2 (*Figure 3—figure supplement 1B*). Near-complete muscle responsiveness may serve as a checkpoint for starting the ecdysis sequence and is possibly implemented by proprioceptive feedback. Class I dendritic arbor (da) neurons, dmd1, vbd, and dbd act as proprioceptors during larval locomotion (*Vaadia et al., 2019*) and remain present at the pupal stage at least through ecdysis (*Figure 3E*). Moreover, bulk $Ca^{++}$

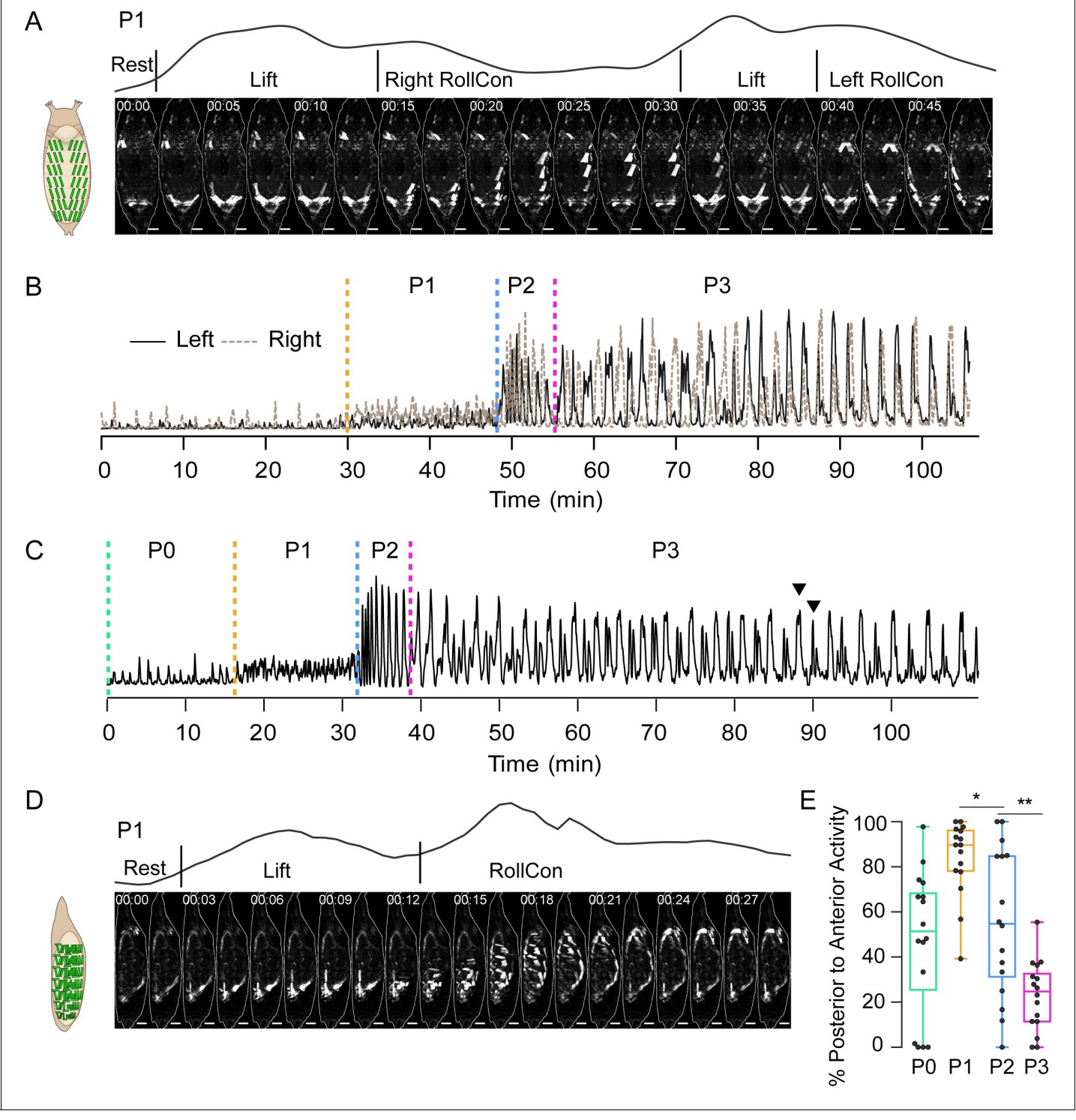

**Figure 2.** Pupal ecdysis muscle activity. (**A**) Muscle activity in a P1 bout (dorsal view) from a pupa expressing *hlk*>GCaMP6s. Schematic on left shows dorsal muscle anatomy. Trace above images shows bulk Ca$^{++}$ activity signal with movements (see **Table 1**). Times in min:s format. Scale bar, 250 μm. T = 0, bout onset. (**B**) Time traces of bulk Ca$^{++}$ activity on the left (black) and right (brown dotted) sides of a pupa executing the pupal ecdysis sequence and imaged from the dorsal side, as in (**A**). Alternating activity is evident. Dotted lines: onset of ecdysis phases. T = 0, imaging onset. (**C**) Time trace of bulk Ca$^{++}$ activity from a pupa imaged from the lateral side. Dotted lines, onset of ecdysis phases. Arrowheads, peak-double peak bouts of late P3. T = 0, imaging onset. (**D**) Muscle activity in a P1 bout (lateral view) from a pupa expressing *hlk*>GCaMP6s. Times in min:s format. Schematic on left shows lateral muscle anatomy. Trace above images shows bulk Ca$^{++}$ activity signal with movements (see **Table 1**). Scale bar, 250 μm. T = 0, bout onset. (**E**)

*Figure 2 continued on next page*

*Figure 2 continued*

Bouts in each phase (%) with P-to-A activity, such as shown in (D). N = 16 pupae. *p<0.05; **p<0.01. See also *Figure 2—figure supplement 1*, *Table 1*, and *Videos 2* and *3* Data presented in this and other figures are available on Figshare DOI: https://doi.org/10.6084/m9.figshare.c.5489637.v1.

The online version of this article includes the following figure supplement(s) for figure 2:

**Figure supplement 1.** Workflow for muscle imaging.
**Figure supplement 2.** Image analysis for muscle activity and body wall movements.

activity in sensory neurons is correlated with muscle activity during P0 (*Figure 3F*), and sensory neurons commonly show correlated $Ca^{++}$ activity with adjacent muscles (*Figure 3G*). When class I da neurons were suppressed with UAS-Kir2.1, $Ca^{++}$ became sustained and widely distributed during P0 before twitching stopped (*Figure 3—figure supplement 2A, B*). Unexpectedly, all animals died before P1 (n = 10).

## Muscle activity patterns identify elementary movements

The transition from P0 to P1 is accompanied by the first overt movements, with bouts typically consisting of a Lift followed by a RollCon. The transition from P1 to P2 is demarcated by a sudden behavioral switch in which a P1 bout is followed by 4–5 bouts containing only a Swing. We used changing muscle activity patterns with associated body wall displacements to define five further canonical movements, all executed in characteristic anatomical compartments (*Figure 4A*, *Video 5*, *Table 1*). We also define a precise onset for P3, which had previously been difficult (e.g., see *Diao et al., 2017*; *Kim et al., 2006*).

Muscle activity in a Swing is coordinated across the dorsoventral axis as it travels anteriorly (*Figure 4B, C*). While initial Swings are rapid, they slow after head eversion (*Kim et al., 2006*) and are then accompanied contralaterally by a movement we call the 'Brace' (*Figure 4A, D*, orange box). The Brace is performed by concurrent contraction of lateral transverse muscles M21–23 and M8 in anterior hemisegments, followed by contraction of these same muscles in posterior hemisegments (*Figure 4D*, lateral view). The Brace begins the shift of activity from P-to-A waves in P1 to A-to-P waves in P3. Onset of P3 is indicated by a movement we call the 'Crunch' (*Figure 4A*). The first Crunch follows the last P2 bout after a relatively long interbout interval and combines ventral contractions in the posterior compartment with dorsal contractions in the anterior compartment. The compartmentalized and complex movements that follow the Crunch include what we call the 'Anterior Compression' (AntComp), 'Posterior Contraction' (PostCon), and 'Posterior Swing' (PostSwing). The first two comprise what have been termed 'stretch compressions' (*Kim et al., 2006*). All of these movements are unique to P3. Each of the elementary movements defined in *Figure 4A* and *Video 5* is associated with the activity of specific muscles, and we trained a CNN to recognize and annotate the movements (*Figure 4—figure supplement 1A*). Using the CNN to measure movement durations (*Figure 4—figure supplement 1B, C*), together with measurements of bout and phase durations (see Materials and methods), we characterized the variability of pupal ecdysis behavior at the level of phases, bouts, and movements.

## Behavioral stereotypy increases with spatiotemporal level of description

The relative stereotypy of the pupal ecdysis sequence can be seen from bulk $Ca^{++}$ traces (*Figure 5A*), but variation exists in the bout and interbout interval durations of all phases (*Supplementary file 2*, *Figure 5B*), and in the bout number of P1 and P2 (*Figure 5C*). Coefficients of variation (CV) were lowest for P2 for all phase and bout parameters examined

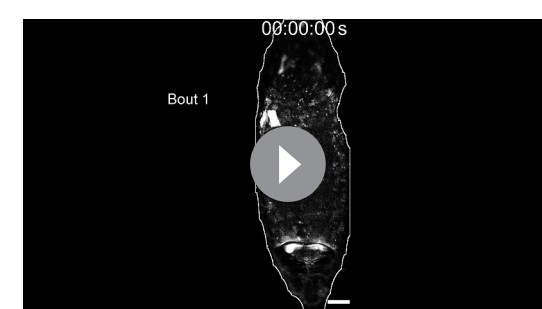

**Video 2.** P1 muscle activity. Dorsal view of a P1 bout with Lift and RollCon movements alternating bilaterally in a *hlk*>GCaMP6s animal; data were sampled at 2 Hz and sped up to 10 fps. Scale bars, 250 μm. Time, seconds.
https://elifesciences.org/articles/68656#video2

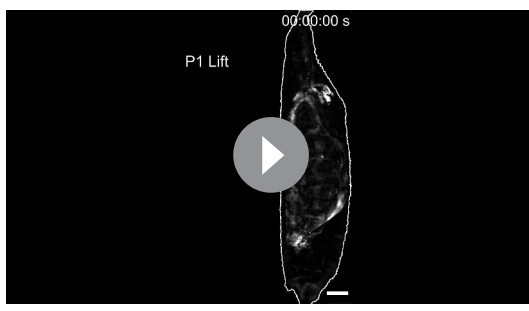

**Video 3.** P1 muscle activity. Lateral view of a P1 bout with Lift and RollCon movements in a *hlk*>GCaMP6s animal; data were sampled at 2 Hz and sped up to 10 fps. Scale bars, 250 µm. Time, seconds.
https://elifesciences.org/articles/68656#video3

(*Supplementary file 2*). This finding is consistent with the developmental importance of P2 and suggests that its execution is the most tightly regulated of all the phases. Movement durations also showed variability, with CVs exceeding 50% (*Supplementary file 2*). However, the order in which movements were executed as determined by the SequenceMatcher algorithm (see Materials and methods) indicated considerable stereotypy. Sequence similarity scores (SS) for movements were computed pairwise for all bouts within each animal by phase and compared across animals. The mean SSs of the sequences for P1, P2, and P3 were all above 0.6, a threshold for similarity (*Figure 5D*), with CVs of 20–44%. P3 had the lowest SS and P2 the highest.

To evaluate the stereotypy of the muscle activation patterns used to generate individual movements, we also used the SequenceMatcher algorithm. The order in which muscles were activated in bouts of P1 yielded an SS of 0.44 ± 0.17, indicating low similarity. However, this SS differed significantly from that of shuffled sequences (0.32 ± 0.12, p<0.001). The sequences are thus not entirely random. Variability was evident between animals (*Figure 6A*) and for individual P1 movements across animals (*Figure 6B*, blue plots). The low similarity of muscle activity patterns suggested that movements may be generated by muscles that are consistently active together even when their individual activation times vary between bouts. Co-active muscle groups have been shown to be important for larval locomotion (*Zarin et al., 2019*); we thus identified groups of muscles for which (1) all muscles were co-active in three or more consecutive frames, (2) the group was identified in at least 80% of the movements in a given animal, and (3) the group was identified in at least 80% of animals. We found eight co-active muscle groups, which we call 'pupal muscle ensembles' (PMEs; *Figure 6D*). Muscles forming a PME are not recruited in a consistent order (*Figure 6C*). In addition, we found several muscles that individually satisfied criteria (b) and (c), but not as part of a group. Collectively with the PMEs, we call these movement-associated muscles 'syllables,' and those active in the eight pupal ecdysis movements are listed in *Table 1*.

We used the SequenceMatcher algorithm to evaluate the stereotypy of syllable activation during movements in each of the three phases (*Figure 6E*). Surprisingly, the order in which syllables were recruited was quite variable, with SSs below 0.5. However, those of P2 and P3 movements were significantly more consistent than the sequence of recruited individual muscles (*Figure 6E*). Syllables associated with P1 movements showed SSs similar to those obtained using the muscle sequences (see also *Figure 6B*, orange plots). This suggests that random muscle activations outside of syllables may contribute to the P1 movements, and such idiosyncratic activations were common in movements of all phases (*Figure 6F*). Because syllables are often confined to particular anatomical compartments, we also compared the activity sequences in the D-V and A-P compartments across P1–P3. P1 bouts exhibit only modest similarity but the bouts of P2 and P3 are intermediate in similarity to those of movements and syllables (compare *Figures 6E* and *5D*). Thus, the observed stereotypy for ordered motor execution in pupal ecdysis is highest for phases, and incrementally decreases with spatiotemporal scale. The least stereotypy (SSs <40%, CVs >40%) is seen in the recruitment order of individual muscles, which is less consistent than the activation of syllables.

## Movements are composed by phasic activation of syllables

The SequenceMatcher results indicate variability in the recruitment of syllables into movements. To more comprehensively examine their contribution to movement, we sought to examine the phasic relationships between syllables that participate in the movements of P1 and early P2 (i.e., prior to the brace). These syllables are shown in *Figure 7A–C* for the lateral, dorsal, and ventral views. The phasic activation of the syllables that can conveniently be monitored from the lateral view during P1 bouts is represented in *Figure 7D*. As expected, these bouts initiate in the posterior compartment (see key, *Figure 7D, E*, right) with the activation of syllables in A6 driving the Lift (*Figure 7D*,

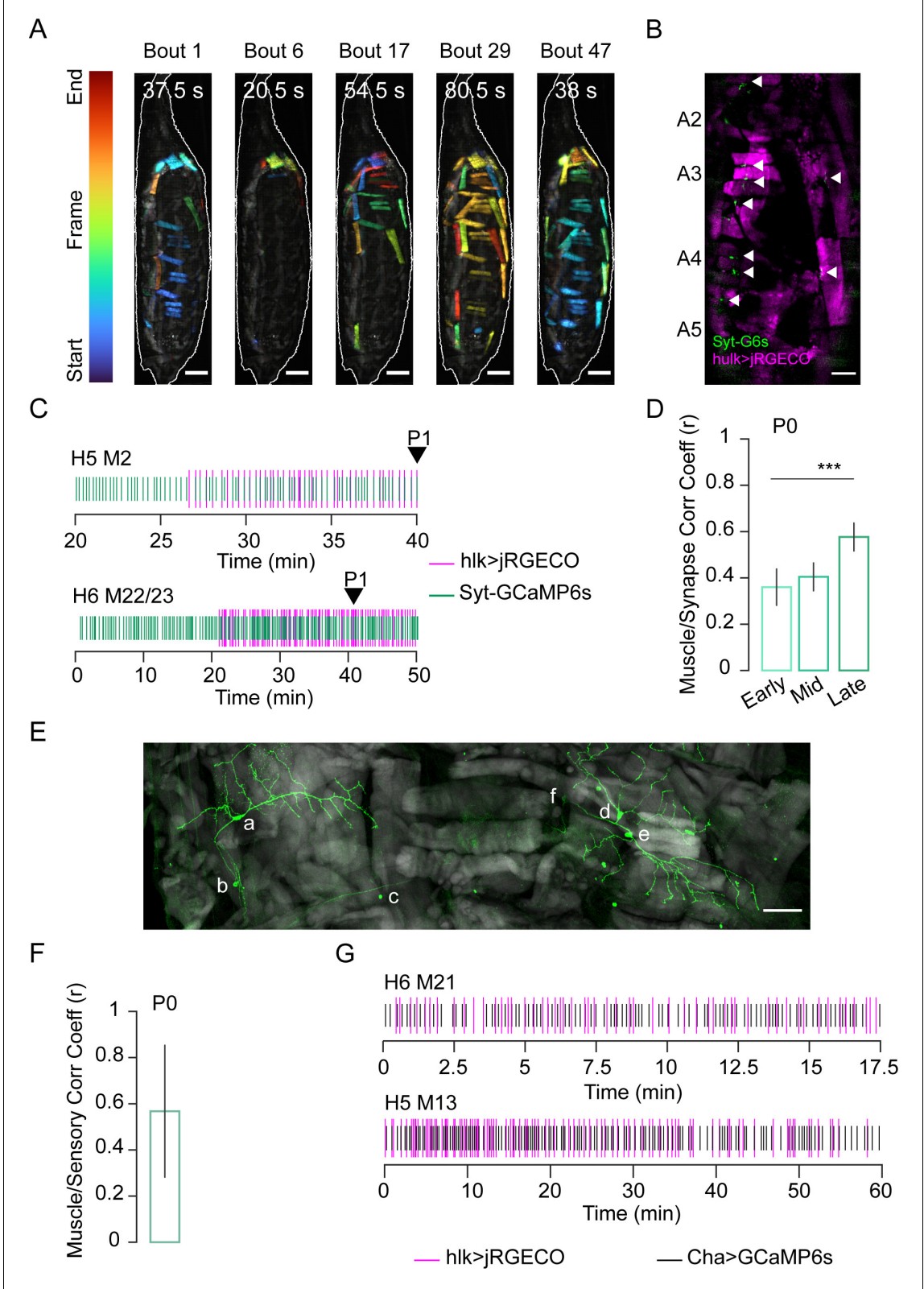

**Figure 3.** Stochastic muscle activity precedes the ecdysis sequence. (**A**) Time-coded projections of muscle Ca$^{++}$ activity (lateral view) in five P0 bouts. Muscle activity is distinct for each bout. Bout durations as indicated; image frames were color-coded according to color scale (left). Scale bar, 250 μm. (**B**) Muscle (magenta; *hlk*>jRGECO) and neuromuscular junction (NMJ, green; *VGlut*>Syt-GCaMP6s) activity in body wall hemisegments (HS) A2–A5. Arrows, active NMJs. Scale bar, 200 μm. (**C**) Representative raster plots generated from peaks in *hlk*>jRGECO (magenta) and *VGlut*>Syt-

*Figure 3 continued on next page*

*Figure 3 continued*

GCaMP6s (green) activity for the indicated muscles and their respective NMJs. Arrow, P1 onset. T = 0, imaging onset. (**D**) Pearson correlation coefficients for *VGlut*>Syt-GCaMP6s and *hlk*>jRGECO activity peaks in multiple muscle/NMJ pairs during early, mid, and late temporal bins, relative to P1 onset. Early P0, N = 19 muscles; mid P0, N = 89; late P0, N = 35. ***p≤0.001. (**E**) HS A3 from pupa expressing mCD8-GFP (green) in class I dendritic arbor (da) neurons using the 410-Gal4 driver (*Vaadia et al., 2019*). Phalloidin-stained muscles (gray). Neuronal somata: a: vpda, b: vbd, c: dbd, d: ddaD, e: ddaE, f: dmd1. Scale bar, 50 μm. (**F**) Pearson correlation coefficient for bulk Ca++ activity peaks in muscles labeled with *hlk*>jRGECO and sensory neurons labeled with *ChaT*>GCaMP6s. N = 10 pupae. (**G**) Rasters compare Ca++ activity peaks of the indicated muscle (magenta; *hlk*>jRGECO) with those in an adjacent sensory neuron (black; *ChaT*-GCaMP6s) during P0. T = 0, imaging onset. See also *Figure 3—figure supplements 1* and *2* and *Video 4*.

The online version of this article includes the following figure supplement(s) for figure 3:

**Figure supplement 1.** Muscle length changes in P0 and M12 responsiveness.

**Figure supplement 2.** Effects of proprioceptor suppression on muscle Ca++.

bottom). This activity precedes the activity of the syllables driving the RollCon in A4 (*Figure 7D*, top). The syllables initially activated in A6 are predominantly located in the ventral compartment and their activity is followed by prolonged activity in the dorsal compartment by PME4, which comprises dorsal longitudinal muscles M1–3. Together, the ventral and dorsal contractions span the P1 bout and serve to compress posterior segments. In anterior segments, compression is transient, with roughly coincident activity of syllables M2 and PME6 in the dorsal and ventral compartments, respectively. This activity overlaps with and is outlasted by contractions of the strictly lateral transverse muscles of PME2 and M8. Contraction of the latter muscles constricts the body wall, effectively pulling the dorsal surface away from the puparium. Separation of the dorsal body wall may be facilitated by reduced surface tension as the RollCons push air anteriorly between the puparium and dorsal body wall. In contrast to P1, syllable activation in P2 bouts is considerably more uniform across hemisegments, body axes, and time (*Figure 7E*). Syllables representing all dorsoventral compartments activate together in each hemisegment, compressing the animal longitudinally and along the D-V axis, with a wave of such compressions traversing the body wall in the P-to-A direction as can be seen by the delayed activation of syllables in A4 relative to A6 (compare dotted lines in *Figure 7E* top vs. bottom). Full details of the compression wave constituting the Swing can be achieved by integration of information about muscle Ca++ activity from the dorsal and ventral views, which permits the reconstruction of the movement from the posterior to anterior end of the animal (*Figure 7—figure supplement 1*). In general, the phasic patterns of syllable activation provide a framework for understanding the evolution of pupal ecdysis movements and behavior.

## Muscle mechanics provide insight into movement composition and function

Although the hemisegmental patterns of muscle Ca++ activity represented by the PMEs provide a general description of the pupal ecdysis movements, not all body wall movement is a consequence of local muscle contractions. This is because the pliable cuticle of the pupa provides little rigidity. Like other animals reliant on a hydroskeleton (*Kier, 2012*; *Kristan et al., 2000*), the pupa uses muscle contractions not only to produce local movement in the body wall, but also to increase hydrostatic pressure of the internal fluid to produce movement in distant parts of the body wall. Isometric contractions of antagonistic muscles also create body wall rigidity to resist body wall distortion so that pressure is appropriately directed. Directing pressure to particular parts of the body is, in fact, a central function of pupal movements, which thus rely on two types of muscle Ca++ activity: activity that results in muscle shortening to deflect the body wall and generate local movement and pressure changes, and isometric activity that does not result in

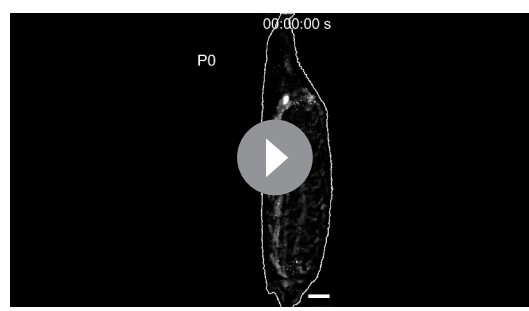

**Video 4.** P0 muscle activity. Lateral view of P0 in a *hlk*>GCaMP6s animal; data were sampled at 2 Hz and sped up to 10 fps. Scale bars, 250 μm. Time, seconds. https://elifesciences.org/articles/68656#video4

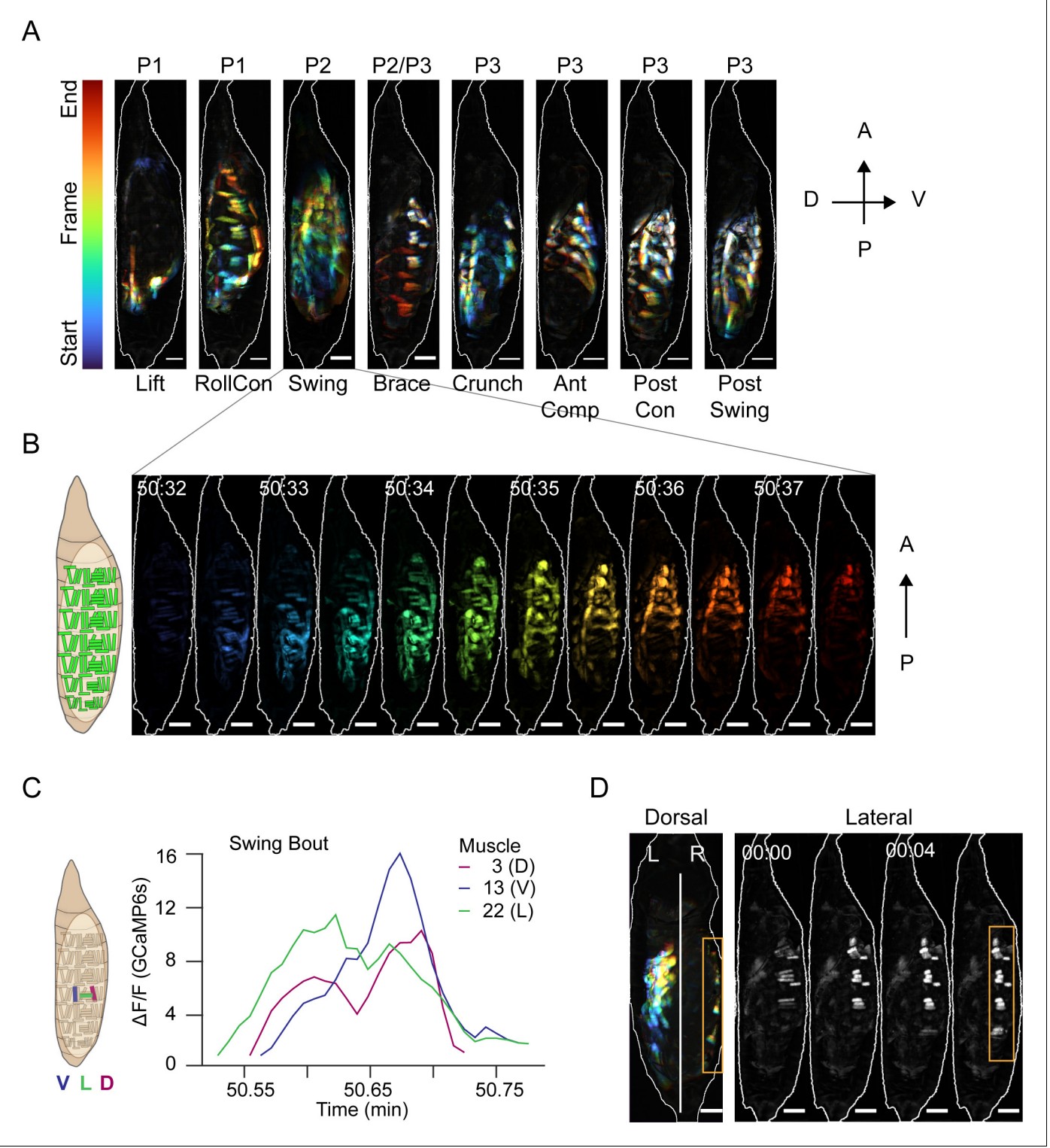

**Figure 4.** Muscle activity patterns identify elementary movements. (**A**) Time-coded projections of muscle activity (lateral view) executed during labeled movements of the indicated phases. Scale bars, 250 µm. (**B**) Muscle activity comprising a Swing from (**A**) color-coded by time showing the P-to-A wave of coordinated activation across the dorsoventral axis. Times in min:s format. Schematic on left indicates muscle anatomy. Scale bars, 250 µm. (**C**) Representative Ca$^{++}$ traces from a single Swing bout measured for dorsal (M3), ventral (M13), and lateral (M22) muscles in hemisegment (HS) A4 show co-incident activity across dorsoventral compartments. (**D**) Dorsal and lateral P2 muscle Ca$^{++}$ activity during the Brace. Orange boxes, active lateral Brace muscles, coincident with Swing movement on the opposite side of the animal (dorsal view). White, dorsal midline. Times in min:s format. Scale bars, 250 µm. T = 0, movement onset. See also *Figure 4—figure supplement 1* and *Video 5*.

*Figure 4 continued on next page*

*Figure 4 continued*

The online version of this article includes the following figure supplement(s) for figure 4:

**Figure supplement 1.** Automated movement detection by convolutional neural network (CNN).

length changes and promotes body wall rigidity to direct pressure.

To better characterize how $Ca^{++}$ activity in muscles of the pupal syllabary generates movement and facilitates and responds to pressure changes, we measured the normalized maximum shortening ($\Delta L/L$) and peak fluorescence intensity ($\Delta F/F$) for each muscle contraction in segments A3–A5 for each ecdysis phase. Increases in fluorescence only moderately correlated with muscle shortening (r = –0.54; *Figure 8—figure supplement 1A*). To determine which muscles shorten the body wall, we calculated the average $\Delta L/L$ for each muscle over each phase, focusing first on P2 because of its role in promoting morphological change. For P2, M12 and the muscles comprising PMEs 1 (M26, M13, M8), 2 (M21–23), 3 (M2, M3), and 4 (M1–M3) shorten the most (*Figure 8A*). As noted above, these contractions create a wave of hemisegmental compressions in the P-to-A direction during the Swing (*Figure 8C*, *Video 5*). The greatest constriction across the D-V axis occurs in posterior segments, consistent with pronounced shortening in PME2 muscles (M21–23) in A5. Progressively decreased shortening of PME2 muscles is observed in A4 and A3. Shortening of the ventral and dorsal longitudinal muscles (M12, M13, M1–3) is more uniform across hemisegments, but the absence of M12 in posterior hemisegments and somewhat greater shortening of the dorsal longitudinal muscles anteriorly is consistent with the greater longitudinal compression of anterior hemisegments (*Figure 8C*).

Comparing $\Delta L/L$ with the corresponding average ($\Delta F/F$) reveals hemisegmental differences (*Figure 8B* vs. *Figure 8A*) including an increase in peak $Ca^{++}$ activity of PME2 muscles (M21–23), moving from A5 to A3 (*Figure 8B*). This trend runs opposite to muscle shortening, which means that in successive anterior hemisegments, the PME2 muscles work harder to generate a smaller length change. This suggests counterforces on the anterior body wall, consistent with increased pressure to evert the head (*Figure 8D*) as has been measured in blowflies (*Zdarek and Friedman, 1986*). Each Swing is initiated posteriorly when the hemolymph is uniformly distributed throughout the body cavity and internal pressure is low. Ascending compression of the body wall on one side pushes the opposite side of the body against the static puparium. This prevents further body wall distension on that side and the compression wave drives hemolymph forward, like squeezing a tube of toothpaste from the bottom up. This creates pressure anteriorly, which is maintained by isometric contractions so that the head is pushed out (*Figure 8E*). While a bilaterally coordinated compression might evert the head more efficiently, the unilateral Swing has a second function: it extrudes the larval tracheal linings on each side of the body. These are deposited in ascending segments on the puparium with each Swing (*Figure 8F*).

The two movements of P1 share features of the Swing and likely prepare the animal for P2. The Lifts draw out the dorsal tracheal trunks, which remain attached to the posterior spiracles (*Robertson, 1936*); the RollCons may help fragment the linings of the stretched trunks in anterior segments so that they are efficiently extruded at P2. Essential to the Lift is compaction of the posterior hemisegments, which is accomplished by bilateral contraction of almost the same syllables as the Swing (*Table 1*). The RollCon, like the Swing, is performed unilaterally in a P-to-A direction. However, it fails to compact hemisegments as it traverses them and only deflects the dorsal body wall. Single-muscle changes in $\Delta L/L$ and $\Delta F/F$ underlying RollCons are much smaller than those for the Swing (compare *Figure 8—figure supplement 1B, C* with

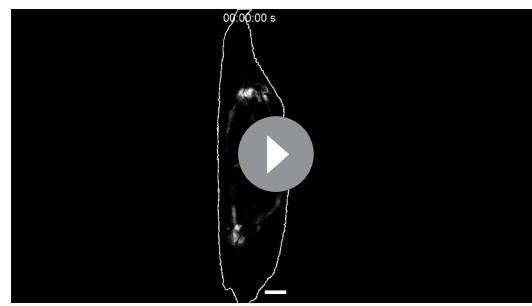

**Video 5.** Muscle activity of the eight canonical movements (lateral view) in a *hlk*>GCaMP6s animal; data were sampled at 2 Hz and sped up to 10 fps, except the posterior contraction, which is 5 fps. Scale bars, 250 µm. Time, seconds.

https://elifesciences.org/articles/68656#video5

**Table 1.** Composition and function of pupal ecdysis movements.

| | P0 | P1 | | P2 | | P3 | | | |
|---|---|---|---|---|---|---|---|---|---|
| Movements | - | Lift | RollCon | Swing | Brace | Crunch | AntComp | PostCon | PostSwing |
| Compartments | - | P D/L/V | A/P D/L/V | A/P D/L/V | A/P L | A/P D/L/V | A D/L/V | P D/L/V | P D/L/V |
| Syllables | | | | | | | | | |
| PME1 | | ✓ | | ✓ | | ✓ | | | ✓ |
| PME2 | ✓ | | ✓ | ✓ | ✓ | ✓ | ✓ | ✓ | ✓ |
| PME3 | ✓ | | ✓ | | | ✓ | | ✓ | ✓ |
| PME4 | | ✓ | | ✓ | | ✓ | | | |
| PME5 | ✓ | | | | | ✓ | | | |
| PME6 | ✓ | ✓ | ✓ | ✓ | | ✓ | ✓ | ✓ | ✓ |
| PME7 | | | | | | ✓ | | | |
| PME8 | | | | | | ✓ | | | |
| M1 | ✓ | | | | | ✓ | | | |
| M2 | ✓ | | ✓ | | | ✓ | | | |
| M8 | ✓ | ✓ | ✓ | | ✓ | ✓ | | ✓ | |
| M12 | ✓ | | | ✓ | | | ✓ | | |
| M13 | ✓ | | | | | ✓ | | | |
| M15 | ✓ | ✓ | | ✓ | | | | | ✓ |
| M26 | | | | | | | | ✓ | |
| L-R rhythm | L-R | - | L-R | L-R | L-R | L-R | - | L-R | L-R |
| A-P rhythm | - | P-A | P-A | P-A | A-P | A-P P-A | A-P | - | A-P |
| Function | - | Fragment trachea | | Evert head, shed trachea | | Elongate appendages | | | |

*Figure 8A, B*). In addition, RollCons engage only a subset of the syllables comprising the Swing (*Table 1*). Rare Ca$^{++}$ activity of M12 is idiosyncratic in P1 and does not contribute to ΔL/L. The coordinated contractions of M12 with other muscles during the Swing, together with the recruitment of additional syllables and higher Ca$^{++}$ activities, explain why RollCons result only in deflections of the dorsal body wall while Swings cause hemisegmental compaction to bend the entire animal (compare *Figure 8—figure supplement 1D* with *Figure 8C*). Overall, the active elements of P1 appear to merge in P2, combining into one robust concerted P-to-A movement.

In P3, coordinated movement across anatomical compartments separates into the multiple compartmentalized movements introduced in *Figure 4A* and elaborated in *Figure 8—figure supplement 2*. These movements form activity patterns without strictly repeating units. Bulk Ca$^{++}$ activity imaged from the lateral side shows an initial oscillatory pattern of variable frequency and amplitude that evolves into a more fixed pattern of alternating wide peaks and slim double peaks (see *Figure 2C*, arrows). The initial variable period of activity is heralded by the Crunch, which is generated by the contralateral activation of PME1 in ventral hemisegments posterior to segment 4 and M2 in anterior dorsal segments. These contractions slightly lift the posterior segments and compact the anterior segments. The Crunch is typically followed by a Brace or an AntComp. The latter movement compacts the dorsal and anterior compartments via contractions of PME7, PME3, and M1 and with ventral activity in PME6. Realignment of the body is achieved by the execution of a PostCon followed by a PostSwing, each composed of syllables in *Table 1*.

The block of movements containing sequentially a Crunch, Brace, AntComp, PostCon, and PostSwing yield an A-to-P flow of activity. They constitute a fairly regular repeating unit with some variation in the order. This block forms the wide peak leading the double peaks seen in the bulk Ca$^{++}$ trace (*Figure 2C*, arrows), and as P3 evolves it increasingly alternates with a modified block that lacks the PostSwing and is followed by long interbout intervals. The double peaks typically consist of

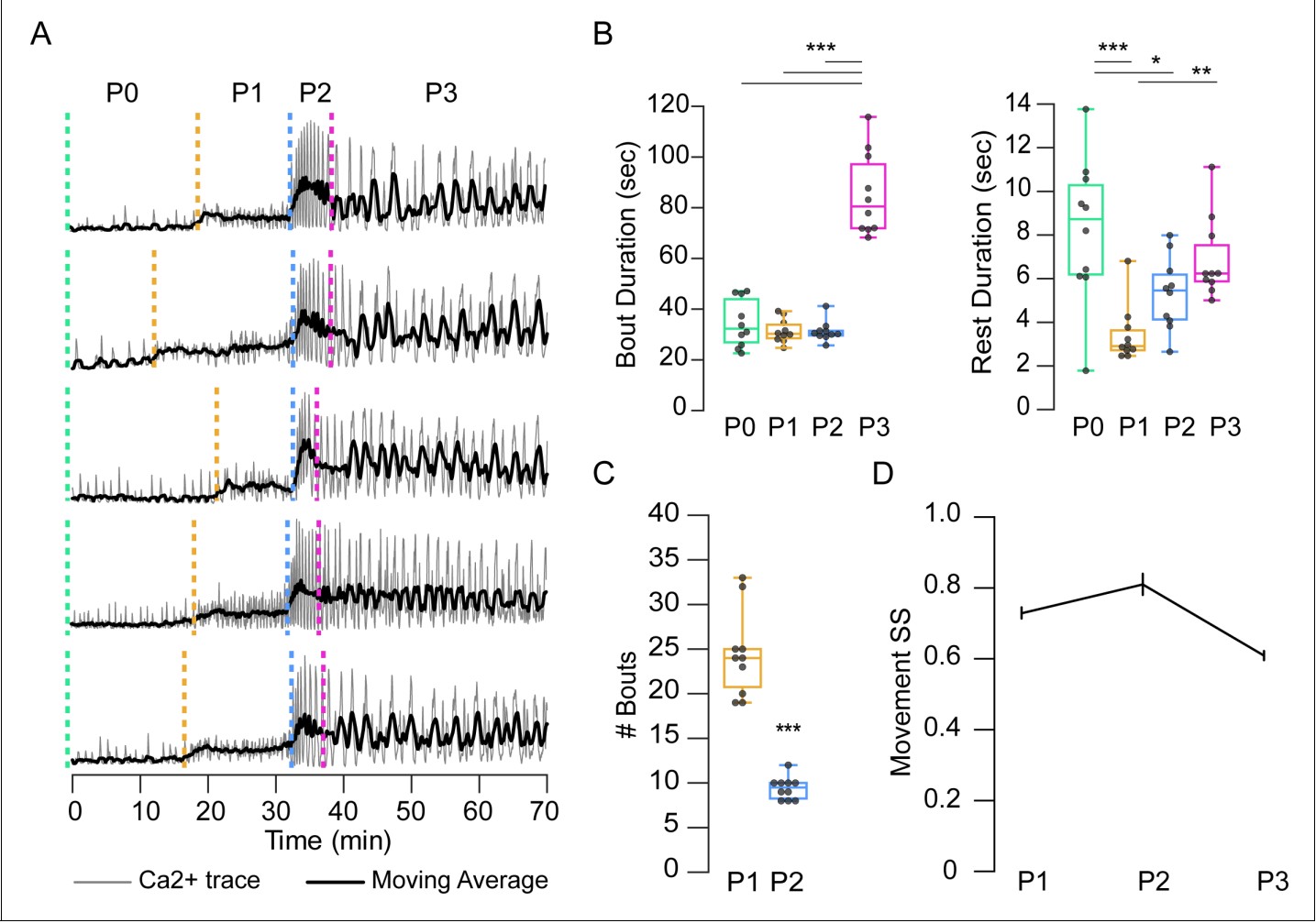

**Figure 5.** Temporal variability in muscle activity. (**A**) Muscle Ca$^{++}$ traces for five representative pupae imaged from the lateral side. Gray, bulk activity; black, 100 frame moving average. Phases as indicated. T = 0, imaging onset, all traces aligned to P2 start. (**B**) Bout (left) and interbout interval (right, 'Rest') durations, for manually annotated pupae (N = 10). *p<0.05; **p≤0.01; ***p≤0.001. (**C**) Number of P1 and P2 activity bouts for pupae in (**B**). ***p≤0.001. (**D**) SequenceMatcher similarity scores (SS) for movement sequences of P1, P2, and P3. See also *Supplementary files 1* and *2*.

a Brace-AntComp-PostCon block followed quickly by a Crunch-Brace combination. In addition, later P3 blocks also include M12 contractions in PostSwings and sometimes Crunches, which visibly increase the compaction of the anterior segments. The increased compaction may increase pressure posteriorly, driving hemolymph into the appendages to lengthen them (*Figure 8—figure supplement 2B, C*).

## Neuromodulators increase behavioral stereotypy

We used the inwardly rectifying K$^+$ channel, Kir2.1, to selectively silence neurons that critically regulate entry into P1 and P2 (*Diao et al., 2016*; *Kim et al., 2015*). Specifically, we suppressed neurons expressing the B isoform of the ETH receptor (N$_{ETHRB}$), and two overlapping populations of neurons expressing CCAP and Bursicon. The former manipulation disrupts P1 initiation by blocking abdominal lifting, while the latter blocks initiation of P2 (*Diao et al., 2017*). We monitored the effects on Ca$^{++}$ activity using *hlk*>GCaMP6s.

In animals in which N$_{ETHRB}$ neurons are suppressed, the baseline increase in bulk muscle Ca$^{++}$ during P1 is severely attenuated relative to WT (*Figure 9A, B*). Although PMEs 2, 3, and 6 characteristic of P1 (*Table 1*) appear 10–15 min prior to P2 (*Figure 9C*), activity in M15 and M1 (and thus PME4) is missing. Muscles of PME1 are also not simultaneously active and thus do not exhibit

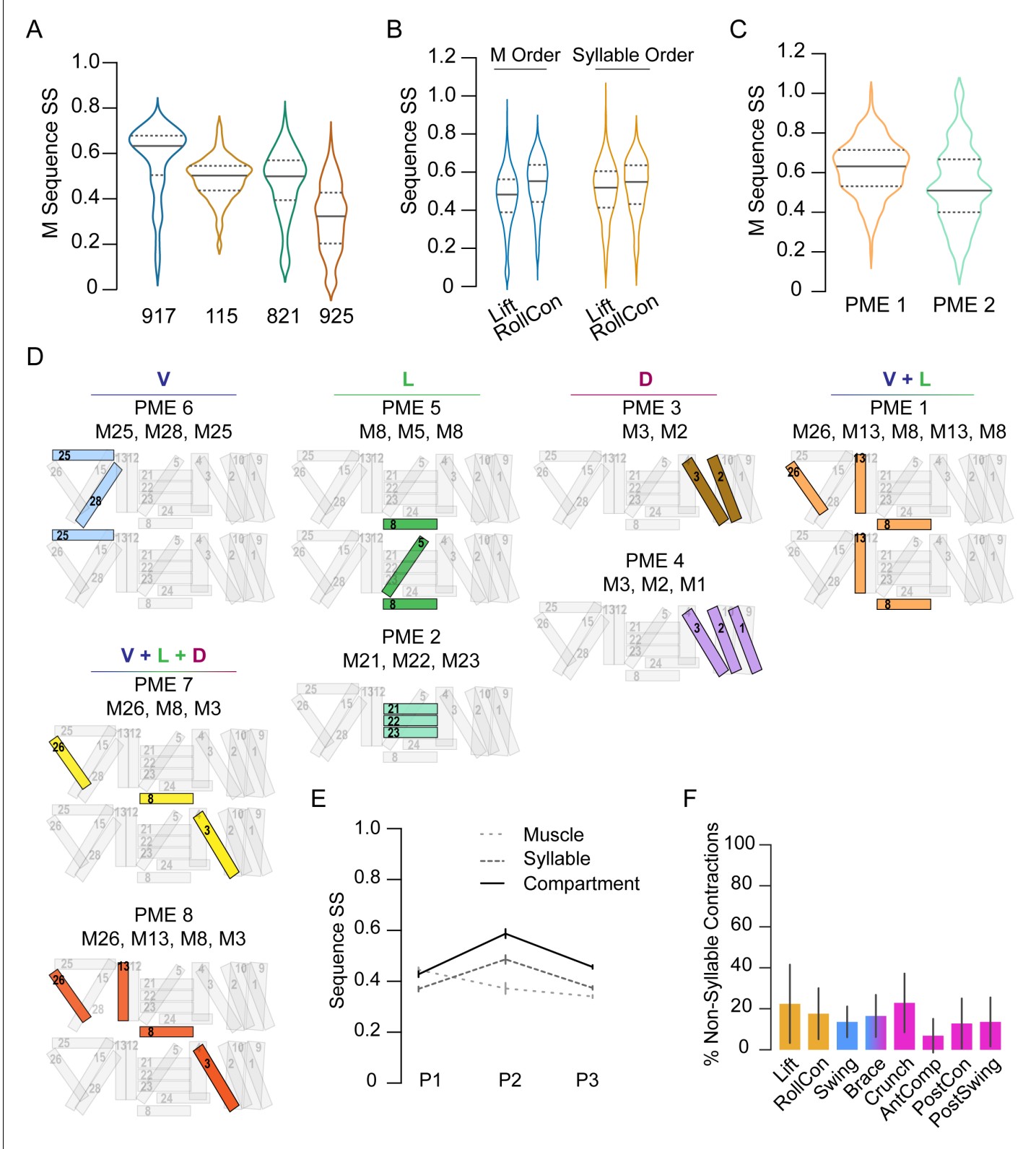

**Figure 6.** Co-active muscles define movement syllables. (**A–C**) Distribution of similarity scores (SS) for single-muscle (M) activation sequences or syllable sequences in four manually annotated *hlk>*GCaMP6s animals, named 917, 115, 821, 925. (**A**) M activation sequences in P1 bouts. (**B**) M activation sequences (blue plots) and syllable sequences (orange plots) in P1 movements across pupae. (**C**) M activation sequences in PME1 and PME2. (**D**) Pupal muscle ensembles (colored) shown schematically on hemisegment musculature and organized by anatomical compartment. (**E**) Mean SS (± SD) for

*Figure 6 continued on next page*

*Figure 6 continued*

activation sequences of compartments (solid line), syllables (dark dashes), and single muscles (light dashes) for each phase by bout. N = 4 pupae. (**F**) Percentage of muscle activations not annotated as part of a syllable, color-coded by phase: orange, P1; blue, P2; pink, P3. N = 4.

ensemble activity (*Figure 9D*). Finally, activity in the D-V compartments is not usually synchronized in the posterior segments. These data indicate that a principal population of ETH-targeted neurons coordinates muscles into syllables to produce the lift movement. Components of the Lift also remain

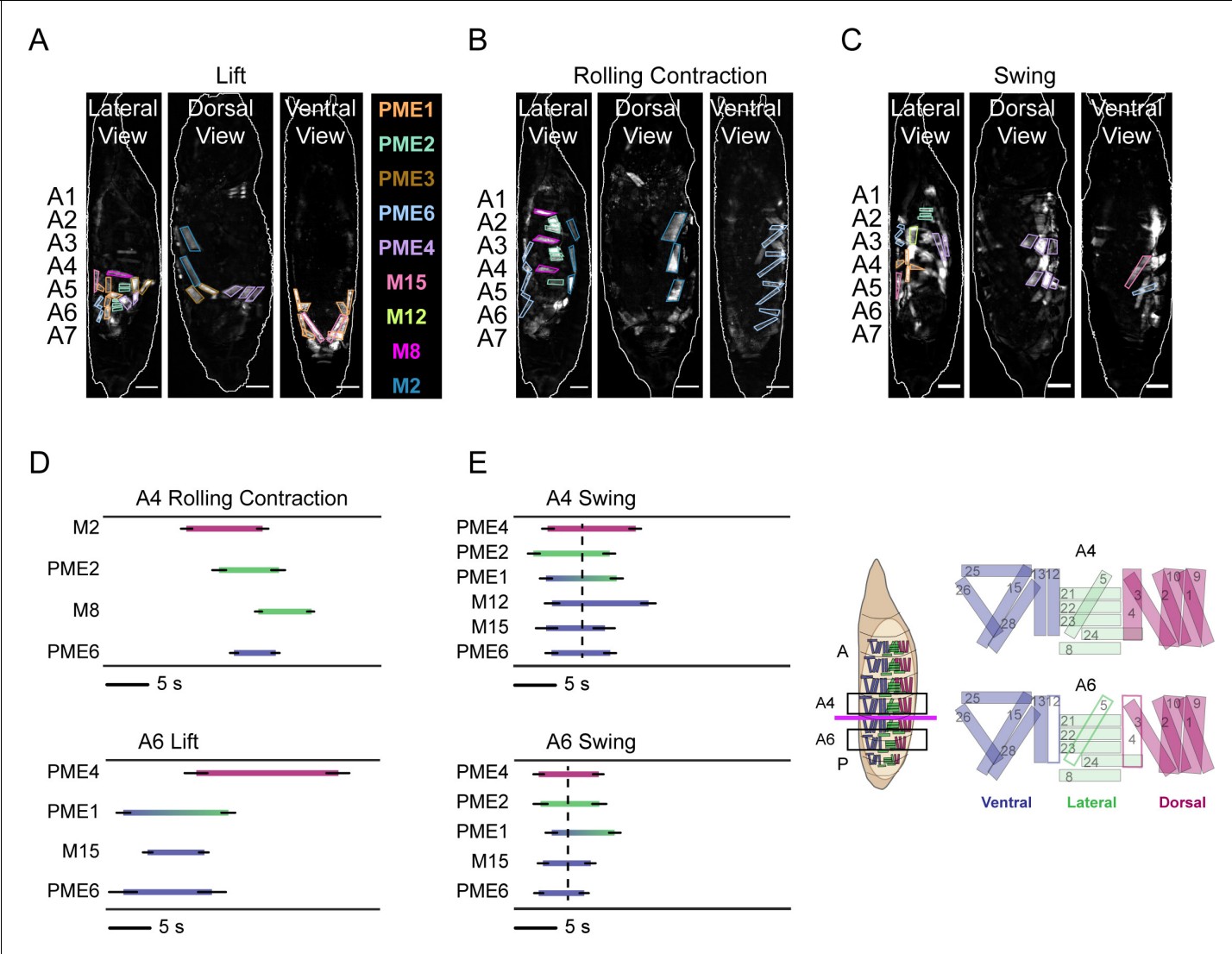

**Figure 7.** Pupal syllable activation drives movement. (**A–C**) Syllables associated with (**A**) Lift, (**B**) RollCon, and (**C**) Swing muscle activity are illustrated in images from the lateral, dorsal, and ventral views. Scale bars, 250 μm. (**D, E**) Phase relationships between syllables visible from the lateral view that form the Lift, RollCon, and Swing in both anterior (A4, top) and posterior (A6, bottom) hemisegments during characteristic bouts of (**D**) P1 and (**E**) P2. The average onset and offset times (± SD; N = 10) are shown, and each syllable is color-coded according to which D-V compartment(s) are occupied by its component muscles. Key at right indicates anatomical compartments: ventral, blue; lateral, green; dorsal, fuschia; and anterior (A)-posterior (P) boundary between A4 and A5 (magenta line). Hemisegments A4 and A6 are boxed and enlarged schematics on right show muscle composition of each. In A6, white muscles are absent. Dotted lines in (**E**) indicate the average midpoint of activation for all syllables for each hemisegment during the P2 bout. See also *Figure 7—figure supplement 1*.

The online version of this article includes the following figure supplement(s) for figure 7:

**Figure supplement 1.** Timecourse of pupal muscle ensemble activation during a swing.

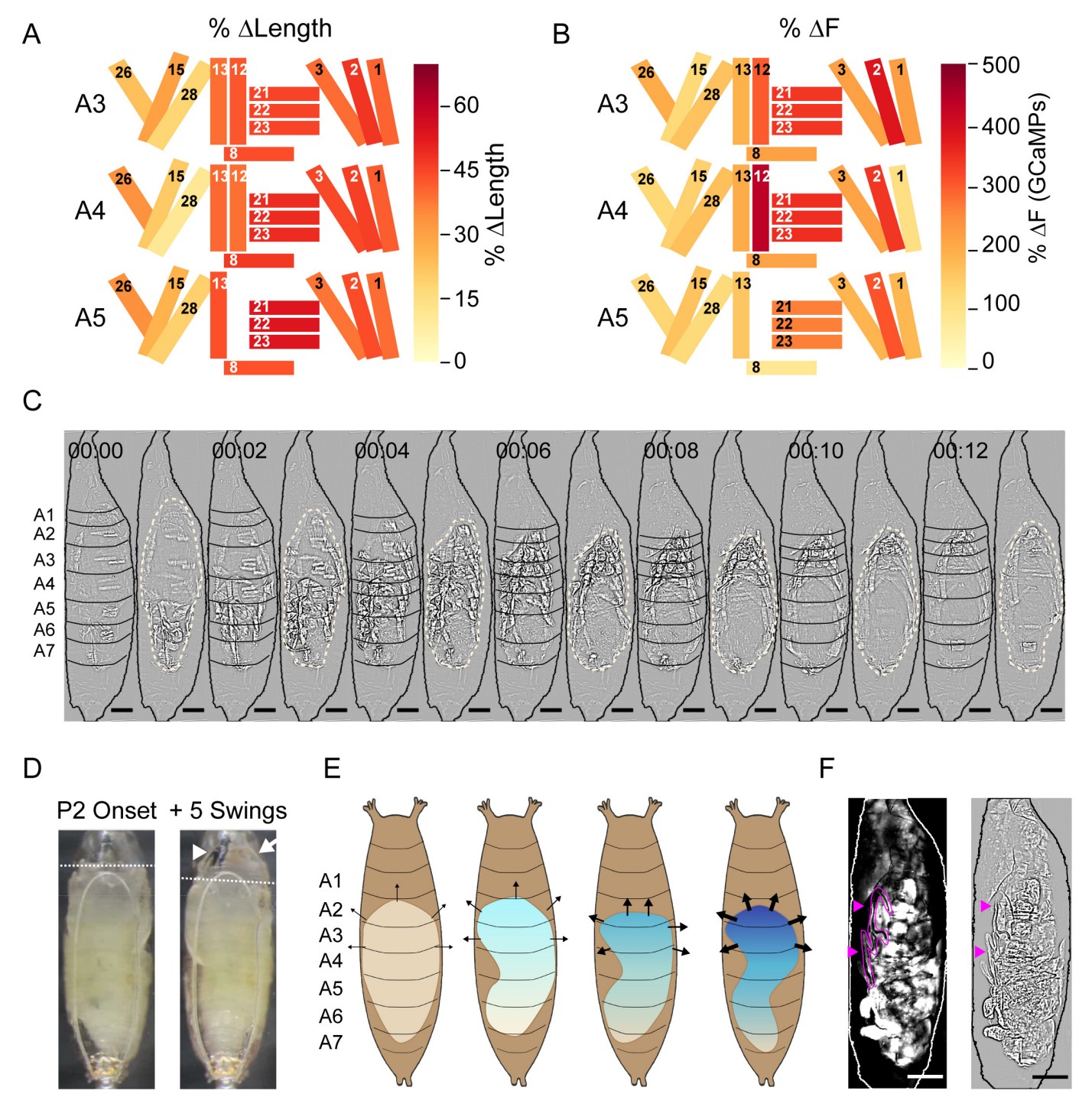

**Figure 8.** Muscle mechanics during P2 and P3 behaviors. (**A, B**) Changes in muscle properties during contraction of indicated muscles in hemisegments (HS) A3–A5 during P2. (**A**) Change in muscle fiber length ($\Delta L/L$). (**B**) Change in GCaMP6s fluorescence ($\Delta F/F$). % changes were calculated from values at activity onset and at maximum activity for N = 24 pupae and color-coded as indicated. (**C**) Muscle activity of a Swing after Laplace transform to show body wall distortion during movement (black, HS boundaries; beige outlines, pupal body). Scale bars, 250 µm. T = 0, movement onset. Times in min:s format. (**D**) Images before (left) and after (right) head eversion. Dotted lines indicate anterior end of the body prior to head eversion. Head, arrow; larval mouth hooks, arrowhead. Scale bars, 200 µm. (**E**) Illustrated effects of the Swing movement on internal pressure. The P-to-A bending of the body wall pushes hemolymph forward, increasing pressure (blue gradient and arrows) in the anterior compartment. Darker blue and thicker arrows, increased pressure. (**F**) Larval dorsal tracheal trunks (magenta arrows) are deposited on the puparium wall during P2 Swings by a

*Figure 8 continued on next page*

*Figure 8 continued*

pupa expressing tdTomato in muscles. Trachea are visualized by scattered light from fluorescent muscles (grayscale, left) and highlighted in magenta in Laplace transformed image (right) for clarity. Scale bar, 250 µm. See also *Figure 8—figure supplements 1* and *2*, *Table 1*, *Video 5*.

The online version of this article includes the following figure supplement(s) for figure 8:

**Figure supplement 1.** Analysis of muscle length and activity changes in P1.

**Figure supplement 2.** P3 movements and appendage extension.

absent from later movements. For example, M1 activity remains disrupted during Swings (*Figure 9— figure supplement 1A*).

Suppressing CCAP-secreting neurons ($N_{CCAP}$) results in normal $Ca^{++}$ activity during P0 and P1, but P2 and P3 activity is absent (*Figure 9E*). Although repetitive P2 swinging is absent, a single, partial swing-like movement is observed after numerous P1 bouts, suggesting that the transition to P2 may be attempted but is not maintained (*Video 6*). M1–3 activate asynchronously in anterior segments so that PME4 fails to form correctly. Consequently, the P-to-A wave on the dorsal side is disrupted by anterior contractions occurring too early (*Figure 9—figure supplement 1B*). The partial swing propagates only through segment A5 and accompanying segmental compression is limited to A6 and A7 (*Figure 9—figure supplement 1C*). The lateral muscles comprising PME2 are unsynchronized with the dorsal and ventral longitudinal muscles (*Figure 9F*). Finally, the dorsal and ventral longitudinal muscles in segments A4–A5 change less in fluorescence ($\Delta F/F = 157.2 \pm 33.2$) and length ($\Delta L = 29.4$ µm $\pm 6.23$) than in WT animals ($\Delta F/F = 272.3 \pm 92.6$; $\Delta L = 42.2$ µm $\pm 2.52$). While $N_{CCAP}$-suppression is lethal (*Diao et al., 2016*), there is persistent activity resembling P1 Lifts and RollCons with no significant reversal in P-to-A direction before death. There are no obvious transitions and our CNN detected few P3-specific movements (*Figure 9—figure supplement 1D*). We conclude that $N_{CCAP}$ modulates several aspects of the transition to P2, including (1) generalized increase in muscle activity during P2; (2) coordination of syllables along the A-P axis that facilitates full body swings; and (3) coordination of activity across the D-V axis, as indicated by the desynchronization of PME2 activity.

The loss of synchronous activity across D-V compartments led us to investigate the innervation pattern of motor neurons that express the CCAP receptor (CCAP-R, *Diao et al., 2017*). Intersectional labeling of CCAP-R-expressing motor neurons using the Split Gal4 system (*Luan et al., 2006*) showed that these neurons innervate approximately half of the pupal muscles via Ib synapses, including all dorsal and three of the six ventral muscles (*Figure 10A, C*). Although none of the motor neurons innervating the lateral transverse muscles express CCAP-R, the transverse muscles M21–23 of PME2 are labeled by the CCAP-R-Gal4 driver (*Figure 10B, C*, *Supplementary file 1*). This suggests that CCAP centrally modulates motor neuron output to dorsal and ventral muscles, while directly modulating lateral transverse muscles. At the larval stage, CCAP is co-released with Bursicon from type III terminals on muscles M12 and M13, which straddle the muscles of PME2 (*Veverytsa and Allan, 2011*). Anti-Bursicon staining established the persistence of type III terminals on M12 (*Figure 10B*, magenta, arrow). In addition, we confirmed the responsiveness of M21–23 to CCAP in fillet preparations treated with bath-applied peptide (*Figure 10D, E*, *Video 7*). Our results demonstrate a role for $N_{CCAP}$ in coordinating syllable activity across both the A-P and D-V axes and a peripheral role for CCAP in directly modulating lateral muscles.

Genetic data suggest that CCAP and Bursicon act synergistically at pupal ecdysis (*Lahr et al., 2012*), and neurons expressing the Bursicon receptor have been identified as essential for ecdysis motor programs (*Diao et al., 2017*). Consistent with Bursicon's colocalization with CCAP in central neurons, we find that suppressing the subset of Bursicon-expressing neurons ($N_{Burs}$) has effects similar to $N_{CCAP}$ suppression: P1 activity is normal, but P2 and P3 are not correctly executed (*Figure 11A*) and the animals die without everting their heads. Animals also execute a single swing-like movement after numerous bouts of P1, but in this case it consists of an entire anteriorly directed wave and some subsequent patterned activity is observed that resembles AntComp, Crunch, Brace, and PostSwing movements, which can be identified by our CNN (*Figure 11B*). This activity lacks organization and remains desynchronized during the swing-like movement activity in the D-V compartments (*Figure 11C*). Additional swing-like movements also occur, but always separated by other types of movement. We conclude that D-V synchronization for abdominal swinging requires $N_{Burs}$, as

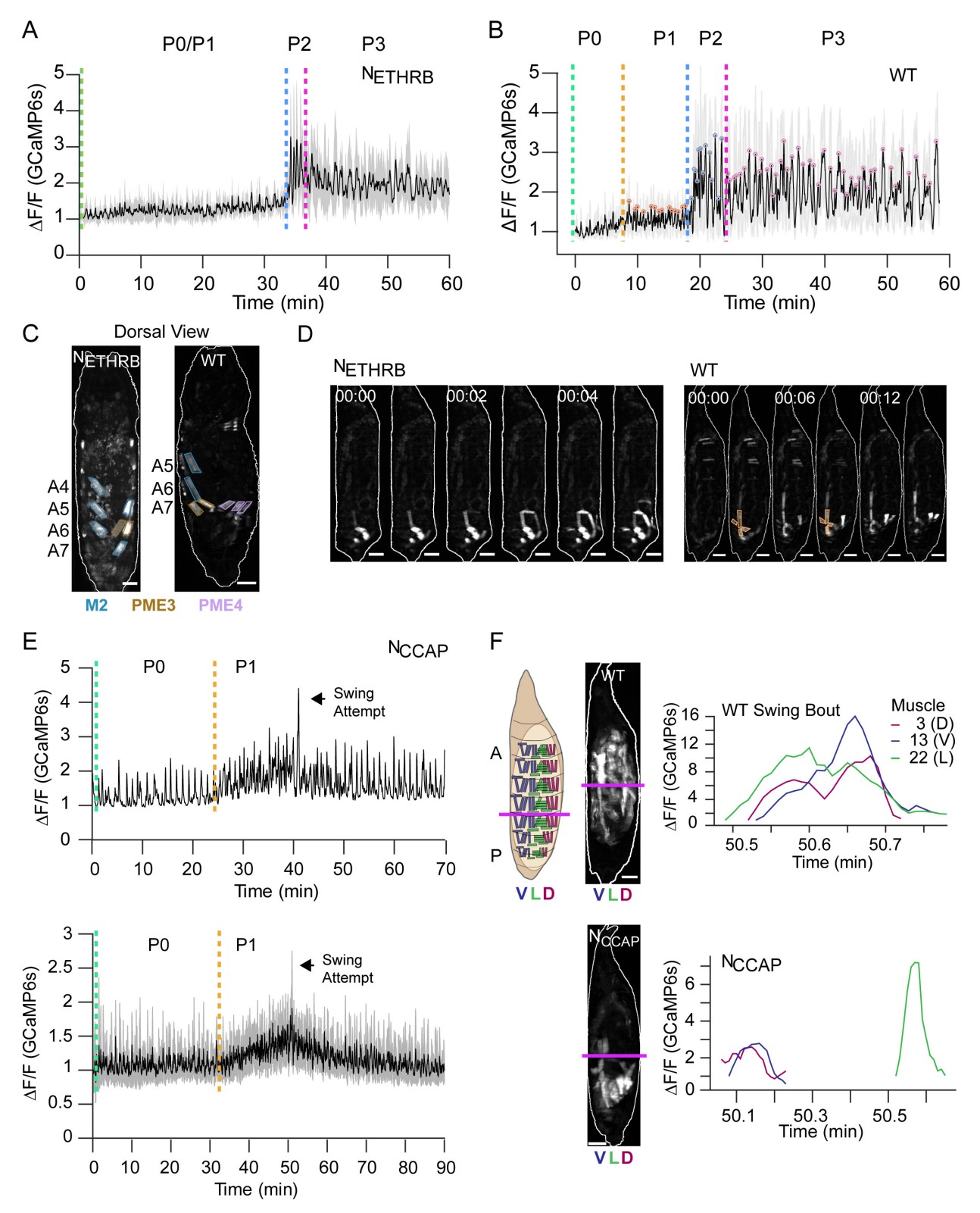

**Figure 9.** Roles of ecdysis triggering hormone (ETH) and crustacean cardioactive peptide (CCAP) in ecdysis behavior. (**A, B**) Lateral view bulk Ca$^{++}$ activity traces, mean (black), and SD (gray) for (**A**) animals with suppressed $N_{ETHRB}$ (N = 6) and (**B**) wild-type (WT) pupae (N = 16). Lack of change in activity precludes discrimination between P0 and P1 for $N_{ETHRB}$-suppressed animals. T = 0, imaging onset, all traces aligned to P2 start. (**C**) Comparison of muscle activity in $N_{ETHRB}$-suppressed ($N_{ETHRB}$) and WT animals during the execution, or attempted execution, of the RollCon movement. Syllables are

*Figure 9 continued on next page*

*Figure 9 continued*

outlined. Scale bars, 250 μm. (**D**) Muscle activity montages for N_ETHRB-suppressed and WT pupae show disruption in posterior muscle activation. PME1 is outlined in WT image. Scale bars, 250 μm. T = 0, movement onset. Times in min:s format. (**E**) Top: Ca$^{++}$ activity trace (lateral view) for an animal with suppressed CCAP-expressing neurons (N_CCAP). Dotted lines indicate identifiable phase onset times. Black arrow indicates a single partial swing-like movement where P2 typically begins. T = 0, imaging onset. Bottom: mean Ca$^{++}$ activity (± SD) for 10 N_CCAP-suppressed pupae. T = 0, imaging onset, all traces aligned to P1 start. (**F**) Images of muscle activity during a Swing in WT or a partial swing-like movement in N_CCAP-suppressed animals. Schematic on left indicates the A-P boundary (magenta) and ventral, lateral, and dorsal compartments. On the right are Ca$^{++}$ traces for M3, M13, and M22 in hemisegment (HS) A4 from a single swing for a WT and N_CCAP-suppressed animal. Co-incidence across the dorsal (D), lateral (L), and ventral (V) compartments is lost in N_CCAP. Scale bars, 250 μm.

The online version of this article includes the following figure supplement(s) for figure 9:

**Figure supplement 1.** Neuromodulatory neuron suppression disrupts pupal muscle ensembles (PMEs).

does sustained execution of swinging behavior, and that full coordination of activity across the A-P axis additionally requires non-Bursicon-expressing neurons in N_CCAP. The results of suppressing neuromodulatory signaling support both central and peripheral roles for the ecdysis hormones in promoting syllable coordination across the A-P and D-V axes. This coordination promotes the observed coherence of behavioral execution despite the variable timing of activation of individual muscles.

## Discussion

Behavior is linked to neural mechanisms by the muscle activity that governs movement. To gain insight into how nervous systems specify behavior, we examined muscle activity during the *Drosophila* pupal ecdysis sequence at single-cell resolution using genetic Ca$^{++}$ indicators. The pupal ecdysis sequence consists of multiple motor programs, dependent for their execution on hormonal cues. We find that hormonal signaling coordinates muscle activity across individual muscle ensembles and anatomical compartments to ensure stereotypy of behavioral execution. Although stereotypy is evident at the level of phases, the recruitment of muscles into movements is not stereotyped and some muscle activity is not correlated with movement. Importantly, a phase of stochastic muscle Ca$^{++}$ activity precedes the onset of behavior, indicating that prior to the action of ETH, nervous system activity exhibits intrinsic variability. This variability is reduced, but not eliminated, by the action of neuromodulators, which incrementally increase behavioral coherence.

### The emergence of stereotypy from variable muscle activity

Our results significantly extend previous descriptions of pupal ecdysis and illustrate the power of pan-muscle Ca$^{++}$ imaging. Behavioral fine-mapping at single-cell resolution permits the definition and automated detection of elemental movements, the identification of a syllabary of movement-associated muscles and muscle ensembles, and the analysis of their sensitivity to neuronal manipulations. Importantly, single-cell analysis permits the identification of muscle activity that is not consistently associated with movements. The most salient example of such idiosyncratic activity occurs in P0, a previously undescribed phase of muscle activity lacking coordinated movement. Variability persists in phases P1–P3, which exhibit idiosyncratic muscle activation comingled with stereotyped movement syllables. Furthermore, muscle recruitment into syllables, and recruitment of syllables into movements, exhibits considerable variability both within and across animals. All observations suggest that variability in the order of recruitment of behavioral elements is a pervasive feature of the pupal ecdysis sequence with stereotypy emerging only at higher levels of behavioral description.

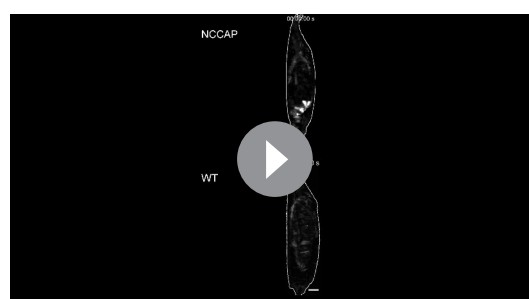

**Video 6.** Partial swing and subsequent activity bouts in an N_CCAP-suppressed pupa (lateral view, top), with wildtype P2 bouts for comparison (bottom); data were sampled at 2 Hz and sped up to 5 fps. Scale bars, 250 μm. Time, seconds.
https://elifesciences.org/articles/68656#video6

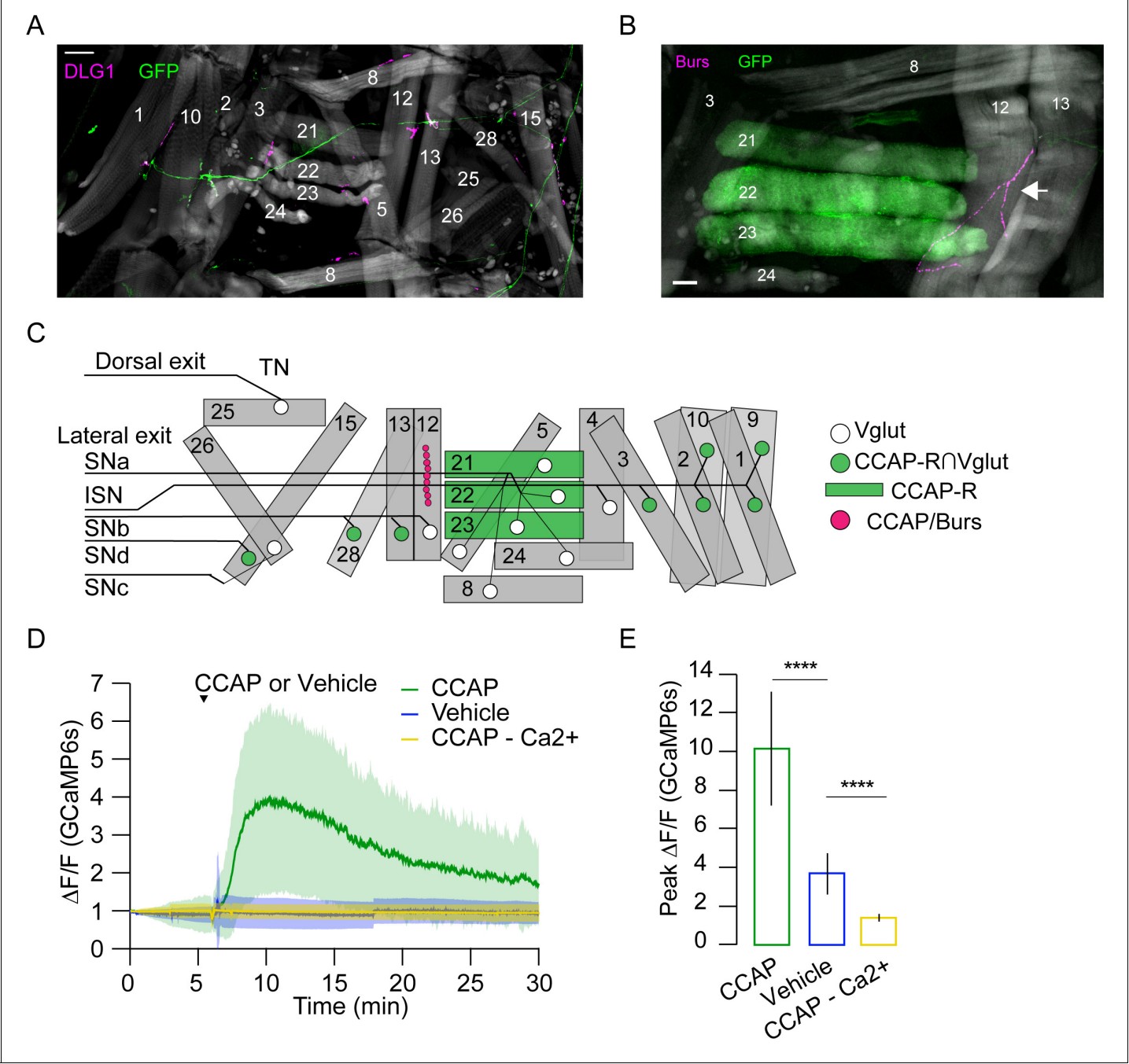

**Figure 10.** Crustacean cardioactive peptide (CCAP) targets a subset of lateral-tranverse muscles. (**A**) Motor axons expressing mCD8-GFP (green) under the control of the CCAP-R∩Vglut Split Gal4 driver innervate a subset of muscles (phalloidin, gray) at synapses stained for the postsynaptic marker DLG1 (magenta). Hemisegment (HS) A3 is shown. Scale bar, 50 μm. (**B**) CCAP-R-expressing muscles M21–M23, visualized with CCAP-R-Gal4>mCD8-GFP (green), are located adjacent to the type III terminal on M12, which is immunopositive for CCAP and Bursicon (magenta, white arrow). HS A4 is shown. Gray, phalloidin-stained muscles. Scale bar, 50 μm. (**C**) Map of CCAP-R motor neuron innervation (green circles) of pupal muscles. Innervation by motor neurons expressing only VGlut, and not CCAP-R (white circles), is also shown, as are muscles expressing CCAP-R (green rectangles) and the type III synapse on M12 that releases CCAP and Bursicon (fuchsia circles). (**D**) CCAP application elicits robust $Ca^{++}$ responses from M21–M23 (green; mean ± SD) in live, filleted *hlk*>GCaMP6s animals. No response is seen with vehicle only (blue line; mean ± SD) or in $Ca^{++}$-free media (yellow; mean ± SD). N = 7–10 pupae, 12 HS/pupa. T = 0, imaging onset. (**E**) Peak GCaMP6s (ΔF/F) activity from experiments in (**D**) expressed as mean ± SD. See also *Supplementary file 1* and *Videos 6* and *7*.

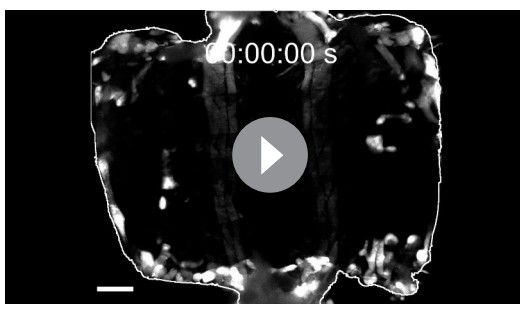

**Video 7.** Response of lateral transverse muscles to crustacean cardioactive peptide (CCAP): live fillet of a hlk>GCaMP6s animal before and after bath application of synthetic CCAP; data were sampled at 2 Hz and sped up to 10 fps. Scale bars, 250 μm. Time, seconds. https://elifesciences.org/articles/68656#video7

Variability in pupal ecdysis behavior may arise from the need to adjust movement to changing forces on the body wall, both from inside and outside. Inside the animal, hydrostatic pressure varies globally in response to local contractions of the body wall. This pressure may need to be countered to maintain control of movement. Outside the animal, the wall of the puparium may form an inhomogeneous substrate as the distribution of molting fluid and air at different places varies. A notable feature of pupal ecdysis is that it is heralded by the appearance of a large air bubble in the abdomen, which is expelled into the puparium by the movements executed during P1 (*Bainbridge and Bownes, 1981*; *Chadfield and Sparrow, 1984*). After expulsion from the body, air is displaced by the animal's movements, first posteriorly and then anteriorly. In the presence of residual molting fluid, pockets of air likely cause fluctuations in surface tension between the body wall and puparium. Forces exerted both by internal pressure and by substrate interactions within the puparium may thus require that motor output be dynamically adjusted, presumably by sensory cues.

The importance of sensory cues to pupal ecdysis is evident from our finding that animals lacking proprioceptive input die at P0 without initiating the behavioral sequence. The cause of death remains to be determined, but its timing suggests that proprioceptive feedback may signal muscle responsiveness to neural input during the period of muscle reactivation and thus provide a readiness signal for ecdysis initiation. Alternatively, pressure on the body wall due to air bubble growth may trigger pupal ecdysis. Sensory cues have been shown to gate behavioral transitions in the adult ecdysis sequences of crickets and also to adjust motor program execution when extrication from parts of the old cuticle fails (*Carlson, 1977*).

For the pupal ecdysis sequence, more work will be required to determine the sources of the observed variability. Notably, sources that arise frequently in the context of other behaviors, such as external environmental stimuli (i.e., stimuli outside the puparium) and competing physiological needs, are absent for pupal ecdysis. In addition, proprioceptive cues, while they may tune ecdysis behavior, are not essential for generating it in that a fictive sequence is generated by an excised pupal brain treated with ETH (*Diao et al., 2017*; *Kim et al., 2006*; *Mena et al., 2016*). Finally, our evidence suggests that at least some behavioral variability derives from the operation of the ecdysis neural network itself since stochasticity is clearly evident at P0 and then appears to extend to other phases.

The variability of P0 muscle bouts is reminiscent in some ways of the neurogenic bursts of muscle activity observed in *Drosophila* embryos prior to hatching (*Crisp et al., 2008*). Initial embryonic bursts consist of uncoordinated muscle activity that becomes increasingly organized over several hours as the locomotor networks mature (*Crisp et al., 2011*). By the time of hatching, complete peristaltic motor sequences are regularly performed. Similar precocious network activity is also found in a variety of other systems (*Blankenship and Feller, 2010*; *O'Donovan, 1999*), including pupal moths where the developing circuitry for flight drives low-threshold neuromuscular responses in muscles that fail to elicit contractions (*Kammer and Kinnamon, 1979*; *Kammer and Rheuben, 1976*). Such activity has also been proposed to support network maturation. P0 motor activity may share this function as patterns of muscle activity become increasingly complex during P0 and recognizable syllables emerge (*Table 1*). However, in contrast to other systems, fully coordinated activity does not appear until the phase ends with the first P1 Lift and even after this muscle and syllable recruitment remain irregular, suggesting continued network variability.

This apparently intrinsic variability may relate to the tradeoffs required of any multifunctional system. Pupal ecdysis, like most behaviors, depends on the integration of signals from CPGs, proprioceptors, neuromodulators, and possibly higher-order command systems—all of which are likely used in the context of larval behavior. For example, circuits that generate waves of activity along the

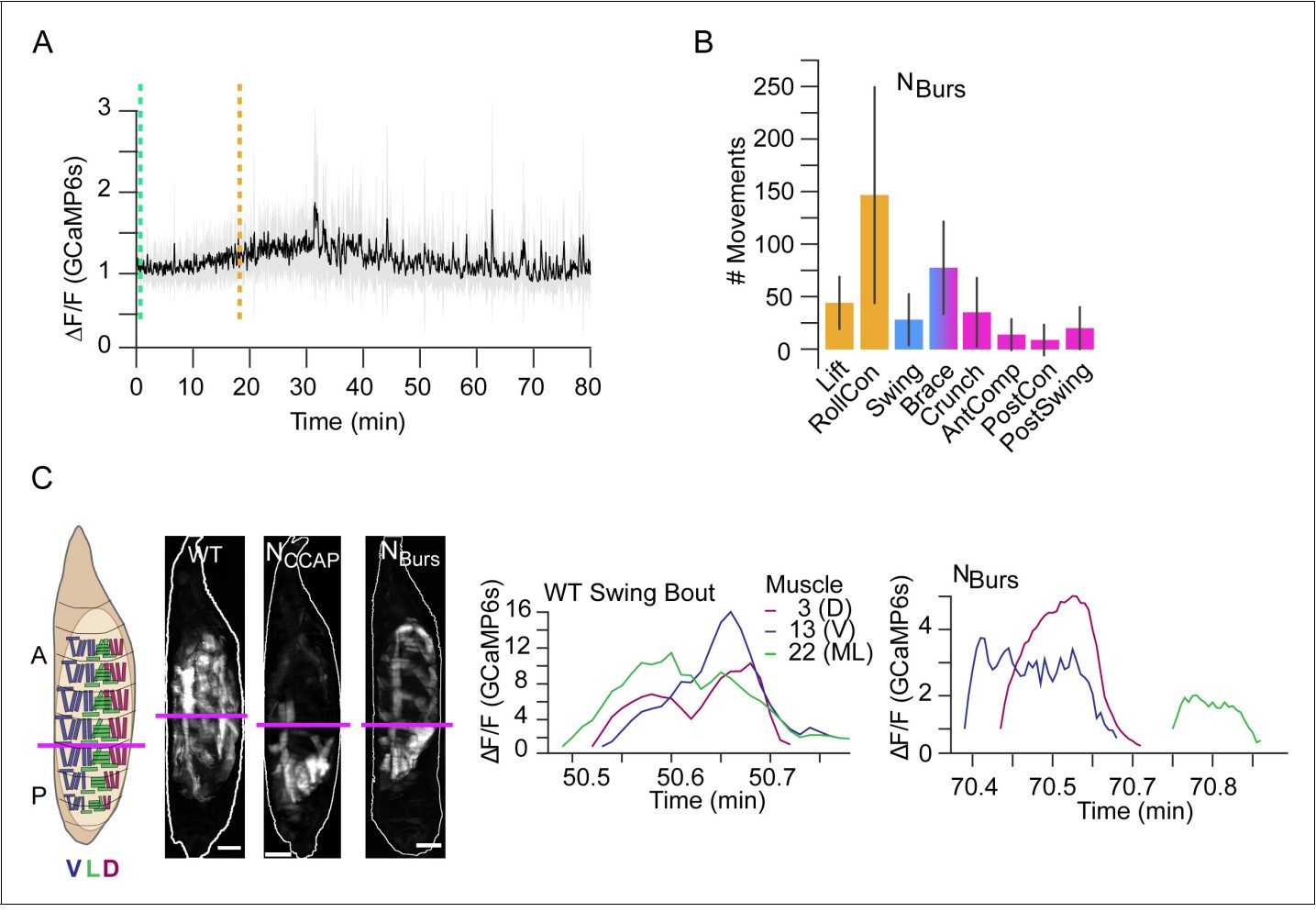

**Figure 11.** Compartmental muscle synchrony for swing requires Bursicon neurons. (**A**) Bulk Ca$^{++}$ activity trace (lateral view) for animals in which the Bursicon-expressing neurons (N$_{Burs}$) are suppressed. Mean activity (black) ± SD (gray) is shown for 10 animals. Dotted lines indicate identifiable phase onset times. P2 consists of a single attempted swing, though subsequent attempts and some P3-like activity follows. T = 0, imaging onset, all traces aligned to P1 start. (**B**) The number of movements as detected by convolutional neural network (CNN) in N$_{Burs}$ animals plotted as mean (±) SD, color-coded by phase: orange, P1; blue, P2; pink, P3. N = 10 animals. (**C**) Images on the left show muscle activity comprising Swing, or swing-like, movements in wildtype (WT), N$_{CCAP}$-suppressed, and N$_{Burs}$-suppressed animals (schematic as in *Figure 9F*); Ca$^{++}$ traces on right show activity of muscles M3, M13, and M22 in hemisegment (HS) A4 from a single Swing bout in a WT animal (left) compared with traces for the same muscles from the swing-like bout in a N$_{Burs}$-suppressed animal (right). As for N$_{CCAP}$-suppressed animals, coincidence across the dorsal (D), lateral (L), and ventral (V) compartments is lost in N$_{Burs}$-suppressed animals. Scale bars, 250 μm.

anteroposterior axis are required for both larval locomotion and pupal ecdysis, but while they generate coordinated bilateral activity in locomotion (*Heckscher et al., 2012*; *Lemon et al., 2015*; *Pulver et al., 2015*; *Zarin et al., 2019*), they generate mostly alternating activity during ecdysis. This degree of flexibility may be bought at the cost of reproducibility of execution. Indeed, precise reproducibility may not be prioritized by nervous systems, which may instead favor solutions that are 'good enough' as has been generally proposed for biological control systems (*Partridge, 1982*). According to this view, further optimization of pupal ecdysis—in the form of behavioral stereotypy—may incur costs on performance in the execution of other behaviors that rely on the same circuit elements.

## Neuromodulatory regulation of behavioral transitions and variability

Variability in pupal motor output is substantially altered by the neuromodulatory action of the ecdysis hormones ETH, CCAP, and Bursicon. Previous evidence indicates that these hormones induce the motor activity characteristic of P1 and P2, even in an isolated pupal nervous system (*Diao et al.,*

*2017*; *Kim et al., 2006*). Consistent with the ability of neuromodulators to reconfigure motor networks (*Marder, 2012*), this induction likely represents reorganization and possibly stabilization of network activity, and also increased network coherence. P1 is distinguished from P0 by the presence of coherent movements and P2 is more coherent than P1 in recruitment of muscles, syllables, and compartmental activity. The reduced stereotypy in P3 may result from waning neuromodulatory action. The ecdysis hormones thus reorganize motor output and increase the stereotypy of its execution. This conclusion is supported by suppression of neuromodulatory signaling, which disrupts muscle activity at multiple levels, as expected from the broad distribution of ecdysis hormone receptors (*Diao et al., 2017*; *Diao et al., 2016*; *Kim et al., 2006*).

What causes the release of Bursicon, CCAP, and ETH at pupal ecdysis is unknown, but our results suggest that ecdysis activity itself may be a factor. Release of ETH occurs only after muscle responsiveness to neural stimulation is ensured, suggesting that the latter may represent a checkpoint for release. Similarly, the aborted swing-like activity occurring in the absence of $N_{CCAP}$ and $N_{Burs}$ activity indicates that entry into P2 has a hormone-independent component that may act as a checkpoint for hormone release, which then sustains P2 network activity. One possible mechanism might be that as the animal pulls back during P1 and creates an anterior space, sensory feedback from the head signals the readiness for P2. Similar checkpoint control mechanisms have been proposed to operate in the adult ecdysis sequence of locusts and crickets (*Carlson, 1977*; *Hughes, 1980b*).

## Neuromodulation and compartmentalization of movement

The granular analysis of muscle activity presented here also reveals how neuromodulators may serve to coordinate action across anatomically and functionally distinct compartmental boundaries. ETHRB neuron suppression blocks the Lift, a movement of the posterior compartment, while suppressing CCAP neurons prematurely terminates the first (and only) swing-like movement by blocking its progression into the anterior compartment. Additionally, the distribution of CCAP-R appears to reflect mechanisms for selectively regulating distinct compartments across the D-V axis. CCAP-R is expressed selectively in motor neurons that innervate dorsal and ventral muscles. These motor neurons have dendrites that occupy similar positions on the myotopic map and share similar synaptic inputs (*Landgraf et al., 2003*; *Zarin et al., 2019*). This distinguishes them from motor neurons that innervate the lateral transverse muscles, which do not express CCAP-R. However, a subset of lateral transverse muscles (i.e., M21–23) themselves express CCAP-R, and the pattern of CCAP-R expression may thus represent a mechanism for synchronizing activity across the D-V axis during the swing movements of P2 when CCAP is released. Notably, muscle synchrony across compartments in posterior segments is lost in the one, partially executed Swing when CCAP-expressing neurons are suppressed (*Figure 9*) and no further Swings are executed.

The putative source of CCAP for the muscles of PME2 are the type III terminals on muscle M12 (*Veverytsa and Allan, 2011*), which is selectively retained in the anterior compartment (i.e., segments 1–4). CCAP-R-expressing muscles in the anterior compartment may thus receive CCAP modulation prior to those in the posterior compartment. This may facilitate the delayed appearance of the contralateral brace, which is formed principally by segmental PME2s and which changes the character of the Swing after head eversion. Interestingly, the anatomical asymmetry in the distribution of M12 is one of several we observed in pupal muscles across the D-V and A-P body axes. These have only been partially described previously (*Liu et al., 2010*) and result from the selective degradation of larval muscles in the ventral and posterior compartments. The asymmetries in muscle distribution correlate with several pupal movements that are physically partitioned across the A-P axis, including the Lift, which occurs only in the posterior compartment during P1, and the AntComp, which is restricted to the anterior compartment during P3. Synaptic mechanisms for controlling movements across the A-P boundary in the larva have been described by *Tastekin et al., 2018*, and similar mechanisms may synergize with anatomical asymmetries and neuromodulatory mechanisms in the pupa to spatially and temporally constrain muscle activity to specific compartments.

It is worth noting that the expression of CCAP-R by the lateral transverse muscles indicates the potential importance of muscle modulation in shaping behavior. Whether modulation of muscle properties also underlies the generalized failure of muscles to respond to synaptic input at the onset of P0 is unclear. Animals stop moving shortly after pupariation and do not resume until pupal ecdysis, but neuromuscular activity may be maintained throughout this period. We observed neuromuscular activity without muscle responses at the earliest pre-pupal stages we examined (approximately

stage P2 of *Bainbridge and Bownes, 1981*; data not shown). However, more detailed imaging would be required to determine the extent, continuity, and pattern of the input. *Drosophila* muscles are targets of a variety of neuropeptides including myoinhibitory peptides, which have also been implicated in pupal ecdysis behavior (*Kim et al., 2015*), and it is possible that these play a role in suppressing muscle responses to neural input. Alternatively, neuromuscular signaling may be progressively potentiated during P0. Such a mechanism has been proposed to operate in pupal moth flight muscles, which respond electrically—but fail to contract—to synaptic input corresponding to flight motor patterns (*Fitch and Kammer, 1986*; *Klaassen and Kammer, 1985*). As animals approach eclosion, rising octopamine levels upregulate neuromuscular efficacy so that flight muscles activate in response to input.

## Identifying neural determinants of behavior

A goal of computational neuroethology (*Datta et al., 2019*) is to describe behavior at a level of resolution that permits the identification of its neural determinants. The muscle-level description provided here lends itself naturally to this purpose. For example, the activity patterns of the muscles comprising PME2 have likely neural correlates within the synaptic and neuromodulatory networks that govern pupal ecdysis behavior. At the neuromodulatory level, the $N_{CCAP}$ cells that terminate on M12 are likely sources of CCAP modulation of PME2 muscles, perhaps to help reverse their P-to-A activation at P2. At the synaptic level, the regular, intersegmental pattern of activation of PME2 muscles in nearly all pupal movements and behavioral bouts (*Table 1*) suggests that PME2 motor neurons are driven by CPG neurons that generate anteroposterior rhythms. Our results thus provide testable predictions about the patterns of neuromodulatory and synaptic connectivity between muscles, motor neurons, and premotor interneurons of various types.

Testing these predictions will be facilitated by data emerging from reconstruction of the larval CNS at synaptic resolution (*Clark et al., 2018*; *Kohsaka et al., 2019*; *Zarin et al., 2019*). Although pupal behaviors differ from those of the larva, broad similarities suggest that at least some neural substrates are shared. The initial P-to-A activity flow of the pupal ecdysis sequence is reminiscent of larval forward peristalsis, and its reversal after alternating bilateral Swing movements resembles the switch to backward peristalsis following sensory stimuli (*Carreira-Rosario et al., 2018*; *Tastekin et al., 2018*). An important difference, however, is that the basic features of larval locomotion are identifiable in the default activity of the excised larval brain (*Lemon et al., 2015*; *Pulver et al., 2015*), whereas the pupal nervous system produces patterned activity resembling pupal ecdysis only in response to ETH. Pupal ecdysis will thus permit investigation of the mechanisms by which intrinsic neuronal activity is organized by neuromodulatory control. Analyzing behavior at single-muscle resolution, as demonstrated here, will facilitate this investigation and should be applicable to other translucent preparations of neuroscientific interest, such as larval fruit flies, roundworms, and larval zebrafish.

## Materials and methods

**Key resources table**

| Reagent type (species) or resource | Designation | Source or reference | Identifiers | Additional information |
|---|---|---|---|---|
| Antibody | Rabbit polyclonal anti-pBurs | Aaron Hsueh/ Willi Honegger | | (1:250) |
| Antibody | Alexa Fluor 555 Phalloidin | ThermoFisher/ Invitrogen | A34055 | (1:500) |
| Antibody | Rabbit polyclonal anti-GFP | ThermoFisher/ Invitrogen | A11122; RRID:AB_221569 | (1:250) |
| Antibody | Mouse monoclonal anti-DLG | Developmental Studies Hybridoma Bank | 4F3; RRID:AB_528203 | (1:500) |
| Antibody | Goat anti-Mouse IgG Alexa Fluor 647 | ThermoFisher/ Invitrogen | A21236; RRID:AB_2535805 | (1:500) |

*Continued on next page*

*Continued*

| Reagent type (species) or resource | Designation | Source or reference | Identifiers | Additional information |
|---|---|---|---|---|
| Antibody | Goat anti-Rabbit IgG Alexa Fluor 488 | ThermoFisher/Invitrogen | A11008 RRID:AB_143165 | (1:500) |
| Peptide, recombinant protein | CCAP | BACHEM | 4015114 | (1 μM) |
| Chemical compound, drug | Schneider's insect medium | Sigma-Aldrich | S0416 | |
| Chemical compound, drug | Schneider's insect medium w/o calcium | Sigma-Aldrich | s9895 | |
| Chemical compound, drug | 2,2-Thiodiethanol | Sigma-Aldrich | 166782 | |
| Chemical compound, drug | ProLong Diamond Antifade Mountant | ThermoFisher/Invitrogen | P36961 | |
| Genetic reagent (*Drosophila melanogaster*) | *D. melanogaster*: ETHRB-Gal4 (ETHRB$^{MI00949}$-Gal4) | *Diao et al., 2016* | | |
| Genetic reagent (*D. melanogaster*) | *D. melanogaster*: CCAP-R-Gal4 (CCAP-R$^{MI05804}$-GAL4) | *Diao et al., 2017* | | |
| Genetic reagent (*D. melanogaster*) | *D. melanogaster*: CCAPR-p65AD (CCAP-R$^{MI05804}$-p65AD) | *Diao et al., 2017* | | |
| Genetic reagent (*D. melanogaster*) | *D. melanogaster*: w; CCAP-Gal4; + | Gift of J. Ewer | | |
| Genetic reagent (*D. melanogaster*) | *D. melanogaster*: VGlut-Gal4DBD (VGlut$^{MI04979}$-Gal4DBD) | *Diao et al., 2015* | | |
| Genetic reagent (*D. melanogaster*) | *D. melanogaster*: UAS-GCaMP6S, insertions on Chromosomes II and III | Bloomington Drosophila Stock Center (BDSC) | RRID:BDSC_42746; RRID:BDSC_42749 | |
| Genetic reagent (*D. melanogaster*) | *D. melanogaster*: UAS-Kir2.1 insertion on Chromosomes II and III | BDSC | RRID:BDSC_6596; RRID:BDSC_6595 | |
| Genetic reagent (*D. melanogaster*) | *D. melanogaster*: yw; UAS-mCD8GFP/Cyo; + | Gift of L. Luo | | |
| Genetic reagent (*D. melanogaster*) | *D. melanogaster*: hlk-T2A-LexA::QFAD | This paper | | |
| Genetic reagent (*D. melanogaster*) | *D. melanogaster*: hlk-T2A-Gal4 | This paper | | |
| Genetic reagent (*D. melanogaster*) | *D. melanogaster*: LexAOp-GCaMP6s, insertion on Chromosomes II and III | BDSC | RRID:BDSC_44589; RRID:BDSC_44274 | |
| Genetic reagent (*D. melanogaster*) | *D. melanogaster*: UAS-jRGECO1a, insertion on Chromosomes II and III | BDSC | RRID:BDSC_64426; RRID:BDSC_63794 | |
| Genetic reagent (*D. melanogaster*) | *D. melanogaster*: 410-Gal4 | BDSC | RRID:BDSC_63298 | |

*Continued on next page*

*Continued*

| Reagent type (species) or resource | Designation | Source or reference | Identifiers | Additional information |
|---|---|---|---|---|
| Genetic reagent (*D. melanogaster*) | *D. melanogaster*: ChaT-Gal4 | *Diao et al., 2015* | | |
| Genetic reagent (*D. melanogaster*) | *D. melanogaster*: UAS-sytGCaMP6s insertion on Chromosome III | BDSC | RRID:BDSC_64416 | |
| Software, algorithm | Illustrator | Adobe | RRID:SCR_010279 | |
| Software, algorithm | Python 3.7 (Anaconda) | Anaconda | RRID:SCR_018317; RRID:SCR_008394 | |
| Software, algorithm | MATLAB | MathWorks | RRID:SCR_001622 | |
| Software, algorithm | LasX | Leica | RRID:SCR_013673 | |
| Software, algorithm | NIS-Elements | Nikon | RRID:SCR_014329 | |
| Software, algorithm | Fiji, ImageJ | NIH | RRID:SCR_002285 | |

## Contact for reagent and resource sharing

Further information and requests for resources and reagents should be directed to and will be fulfilled by Benjamin White (benjaminwhite@mail.nih.gov).

## Experimental model and subject details

### *Drosophila* stocks and rearing conditions

Vinegar flies of the species *Drosophila melanogaster* were used in this study. Flies were raised on cornmeal-molasses-yeast medium or Nutri-Fly German food (Genesee Scientific) and housed at 25°C and 65% humidity in a 12 hr light/dark cycle. Both males and females were used in this study and all experiments analyzed animals at stages L3 (third instar), P2 (*Bainbridge and Bownes, 1981*) ~6 hr after pupariation, or at the time of pupal ecdysis, ~12 hr after pupariation. Fly stocks described in previous publications include VGlut-Gal4DBD (i.e., VGlutMI04979-Gal4DBD) and ChaT-Gal4 from *Diao et al., 2015* and CCAP-R-p65AD and CCAP-R-Gal4 (*Diao et al., 2017*), ETHRB-Gal4 (*Diao et al., 2016*). *hlk* was identified by the larval muscle expression pattern of the NP3137-Gal4 enhancer trap line whose P{GawB} element is inserted at the 5′ end of the *l(2)01289* gene. To make *hlk-T2A-LexA* and *hlk-T2A-Gal4* lines, either PBS-KS-attb1-2-PT-SA-SD-1-T2A-LexA:QFAD-Hsp70 or PBS-KS-attb1-2-PT-SA-SD-1-T2A-Gal4-Hsp70 was injected into y w; l(2)01289$^{MI05738}$/Cy; + embryos (Bloomington stock# 42105) and adults were screened for loss of the y+ marker. All injections were made by Rainbow Transgenic Flies, Inc (Camarillo, CA). The CCAP-Gal4 line was a kind gift from J. Ewer (Universidad de Valparaiso, Chile). All other fly lines, listed in the Key Resources tTable, were obtained from the Bloomington Drosophila Stock Center at Indiana University.

## Method details

### Immunohistochemistry

Third instar larvae and stage P4 pupae with an air bubble and visible gut movement indicating ecdysis proximity (*Bainbridge and Bownes, 1981*) were isolated from their vials, chilled for 20 min on ice, and placed into a dissection dish with 1× phosphate-buffered saline (1× PBS). Animals were pinned dorsal-side or ventral-side up (as indicated) at the anterior and posterior ends, a small incision was made along the entire dorsal or ventral midline, and the visceral organs were removed. Tissues were then fixed in 4% paraformaldehyde for 30 min at room temperature (RT) and washed in 1× PBS three times. A block was performed for 1 hr at RT in PBT (1× PBS, 0.5% Triton-X) with 5% Normal Goat Serum (NGS). The tissues were then incubated with Rabbit anti-GFP (Life Technologies,

USA) diluted to 1:250 in PBT and mouse anti-DLG (Developmental Studies Hybridoma Bank, Iowa City, IA) at 1:500 for 48 hr at 4°C. After six 30 min washes in PBT with shaking, samples were incubated with secondary antibodies Alexa Fluor 647, 488, and Alexa Fluor Phalloidin 555 (Life Technologies) at 1:500 in PBT with 5% NGS for 48 hr at 4°C (Life Technologies). Samples were washed again three times for 20 min washes in 1× PBT with shaking and mounted on #1.5 glass coverslips with Prolong Diamond (Life Technologies). Confocal imaging was performed using a Leica SP8 with AOBS, bidirectional resonant scanning, and a 20×/0.75 NA air objective. Unless otherwise noted, the images presented are maximum intensity projections (MIPs), produced in Fiji ImageJ (*Schindelin et al., 2012*), of multi-point Z-stacks collected through the entire preparation.

## Manipulation of neuronal activity

All neuronal suppression experiments were conducted using two copies of UAS-Kir2.1 in a parental line generated by combining insertions on chromosomes II and III with *hlk*-LexA::QFAD and Lex-AOP-GCaMP6s, respectively. Control animals bearing the UAS-Kir2.1 transgene, but no Gal4 driver, perform the ecdysis sequence normally, and animals in which the 410-Gal4, CCAP-Gal4, ETHRB-Gal4, and Burs-Gal4 drivers express mCD8-GFP are healthy and viable to adulthood.

## Live fluorescence microscopy

### Intact animals

P4 stage pupae were isolated as for immunohistochemistry, rinsed with 50% bleach for 3 min, rinsed in PBS, and immersed in a custom-chamber with 2,2-thiodiethanol (TDE, Sigma-Aldrich), which rendered the puparium transparent and immobilized the animal with the dorsal, ventral, or lateral side facing the objective (*Figure 2—figure supplement 1*). Muscles only on the selected side were imaged using an epifluorescence stereomicroscope (Nikon SMZ25) with a 1×/0.3 NA objective, GFP filter cube, 4× magnification, low depth-of-field, and a partially closed aperture diaphragm. Fluorescence from muscles opposite the imaging field is effectively excluded because the pupal musculature lies superficially along the body wall and light is scattered by internal tissues. All muscle activity on one side could be rapidly captured in a single frame using a high-speed sCMOS detector and data were collected at 2 Hz for 90–120 min.

## Filleted animals

P2 stage pupae (*Bainbridge and Bownes, 1981*) were cold-anesthetized, washed in PBS, and dissected as for immunohistochemistry. The brain was removed by severing ventral nerve cord (VNC) projections and removing the brain and VNC. Once filleted, the PBS was replaced with 1 mL of Schneider's insect medium (SIM, Sigma-Aldrich). Imaging was performed for 30 min on a Nikon SMZ25 stereomicroscope with a 1×/0.3 NA objective and sampled at 2 Hz. After the first 5 min, 1 μM CCAP solution in SIM was added to the bath, for a final effective concentration of 0.5 μM, and image collection continued for the remaining 25 min. The vehicle control was SIM without CCAP. $Ca^{++}$-free controls were conducted with SIM without $Ca^{++}$ (Sigma-Aldrich). CCAP (BACHEM, Torrence, CA) stock solutions were prepared by dilution to 1 mM in water.

## Dual-channel imaging

For experiments requiring GCaMP6s and jRGECO1a fluorescence, animals were prepared as intact animals above (pupal stage P2 for NMJ/muscle experiments and pupal stage P4 for sensory/muscle experiments) and imaging was conducted on a Nikon Ti epifluorescence microscope with a 10×/0.5 NA air objective, Cairn twin-cam, two PCO edge 4.2 sCMOS cameras, and filters for GFP and RFP. Images were acquired at 2 Hz for 60–90 min.

## Image processing

Unless otherwise noted, all live experimental image series were background subtracted in Fiji ImageJ2 (*Schindelin et al., 2012*). Where values of length and fluorescence are indicated, the line tool (width, 3 px) was used to manually measure intensity and length for muscles 1, 2, 3, 5, 12, 13, 15, 22, 26, and 28 in hemisegments 3, 4, and 5 for each of the ecdysis sequence phases at the onset of muscle activity (visible fluorescence), the peak of the muscle activity (brightest fluorescence), and the offset of muscle activity (fluorescence returns to baseline). Single-muscle identification was

facilitated by the histolysis of half of the larval musculature and the occlusion of opposing side muscles by thick and highly scattering internal tissue. Data were collected sub-saturation but most videos and figures presented are contrast-enhanced to aid the eye. Bulk $Ca^{++}$ traces were extracted from ROIs defined as the entire animal excluding the puparium, or as single muscles/neurons where indicated, and normalized to F0, defined here as the average fluorescence intensity over the first 50 frames in the image series. Raster plots were generated from activity peaks identified from $Ca^{++}$ traces using a peak finding algorithm in Python with manually determined thresholds for GCaMP6s and jRGECOa1.

## Behavior annotation

A subset of four animals from the total *hlk*>GCaMP6s experimental dataset (N = 16) were annotated frame-by-frame by expert raters to identify the muscles newly activated in each frame. These data were used to determine the frequency and order of activation for each muscle. Movements were annotated manually for these four animals plus one more by visual inspection of muscle activity pattern and deflections of the body wall with respect to the puparium from raw image series (before background subtraction) that were contrast-enhanced to allow visualization of autofluorescence from the body wall. Bouts were identified manually from a subset of 10 *hlk*>GCaMP6s animals as the start of a sequence of muscle activations flanked on either side by $\geq$ 2 frames of no new activations, which were defined as inter-bout intervals. The criteria for defining PMEs were determined empirically based on frame-by-frame analysis of 10 complete image series of the pupal ecdysis sequence. Groups of muscles that were repeatedly observed ($\geq$80% of bouts in $\geq$8 animals) to be co-active in a regular pattern were subsequently analyzed for the duration of co-activity. A group in which the activity of all muscles overlapped for three or more frames was defined as a PME. Post-hoc analysis revealed that three frames on average represent approximately 15% of the total period of co-active duration for PMEs.

## Quantification and statistical analysis

### Sample calculations and general statistics

Sample sizes were predetermined using power calculations with 85% power from ecdysis phase 2 behavior duration data presented in *Diao et al., 2017*. N calculated for suppression of the CCAP, non-ETHRA population vs. WT is 6 animals and N calculated for suppressing CCAP/ETHRA population vs. WT is 12 animals. Sample sizes in the current paper are listed with each figure and in *Supplementary file 1* and range from 10 to 16 with the exception of manually annotated data, for which sample sizes were not statistically predetermined. Statistical analyses were performed using GraphPad Prism 8 or Python 3.7 (https://anaconda.com, versions 2–2.4.0). Individual statistical methods and parameters are reported in the figure legends.

## Directionality analysis

To find direction of activation (A->P or P->A) (*Figure 1K*), we first computed a MIP along the x (horizontal) axis of the *hlk*>GCaMP6s image. For every frame, the MIP was a 1D vector of length H for an HxW image. Then we computed the location of the mode of the MIP vector on the y axis. An intensity mode indicates the location of the most dominant muscle activation. Therefore, the locations of the modes give an estimate of the direction of motion during the muscle activity period. Note that we are interested in finding the location of maximum motion, which is efficiently captured by the mode of the intensities, not by the mean. Once the intensity modes were identified for every frame using MIPs, we computed the number of modes above and below a threshold, indicating P->A and A->P motion, respectively. The threshold is automatically computed as the median of all modes. We then calculated the ratio of the number of modes below the median line to the number of modes above the median line and used this ratio to determine if the direction of activation is posteriorly or anteriorly directed.

## Automated movement analysis

We used a CNN to automatically estimate the type of motion in each *hlk*>GCaMP6s image frame. The network model is a modified version of Inception-v3 (*Szegedy et al., 2015*). To fit the model in available GPU memory based on the image size, we removed the final Dense layer and replaced it

with a GlobalAveragePooling2D layer. In addition, a dropout layer was also added to introduce stochasticity into the model to avoid overfitting. There were totally 21.8M parameters in the model. The detailed network is described in the train.py provided as supplementary material. The model was trained to predict nine motions (Anterior Compression, Posterior Swing, Brace, Crunch, Rolling Contraction, Lift, Posterior Contraction, Rest, Swing) from the *hlk*>GCaMP6s images of five animals. For each animal, the images were resized to 403 × 129 × N, where N is the variable number of frames based on the duration of the phases. Each frame was obtained at 2 Hz. Every frame, starting from phase 1, was assigned to one of the above-mentioned motions and was derived by consensus of three expert raters. Since the motion cannot possibly be derived by looking at a single frame, we used windows of 25 frames (12 previous and 12 following) to estimate the motion at one frame. Therefore, the training data for each frame consisted of a 403 × 129 × 25 matrix. The CNN model was then trained by predicting the motion of the center (13th) frame.

Since there were only five animals available for training, we used all possible overlapping frames to augment the training data. Categorical cross-entropy with the Adam (*Kingma and Ba, 2015*) optimizer was used with learning rate of 0.0001 to obtain a probabilistic estimation of motion for each frame. An early termination criterion was used to prevent the model from overfitting, where the training was stopped if the categorical cross-entropy did not increase for 10 consecutive training epochs. *Figure 4—figure supplement 1A* shows the accuracy of motion prediction averaged over all available frames for each of the five training animals in a leave-one-out cross-validation. The final (magenta) curve shows the training accuracy where frames from all five animals were aggregated and then the model was trained on the aggregated frames. Similar accuracy between cross-validation and aggregated training shows that the CNN model did not overfit the data. However, to use information from all training animals, we applied the aggregated trained model (i.e., magenta) on the remaining 11 animals. For every frame of a test animal image, the motion with maximum probability was used as the final motion. For better accuracy, the predicted motions of the 11 animals were corrected by an expert rater.

## Muscle activation order analysis

The annotated muscle datasets were sorted by onset frame number and then by muscle. Then, cells were concatenated into strings by bout, movement, or syllable, and assessed pairwise via the Python SequenceMatcher algorithm to score for similarity in string order. Other behavioral features were similarly analyzed using the onset times of syllables, movements, or compartments to order the sequences. Documentation for this algorithm is provided by the difflib library (https://docs.python.org/3/library/difflib.html).

## Acknowledgements

We thank Dr. Feici Diao for advice and help initiating this project and the three reviewers and reviewing editor for their insightful suggestions. We are grateful to the Bloomington Drosophila Stock Center (NIH P40OD018537) for fly lines used in this study and to Dr. Ted Usdin and the NIMH Systems Neuroscience Imaging Resource (ZIC-MH002963) for use of the Leica SP8 confocal microscope. This work was supported by the Intramural Research Programs of the NIGMS, NIMH, NIDDK, and the NIBIB.

## Additional information

### Funding

| Funder | Grant reference number | Author |
| --- | --- | --- |
| National Institute of General Medical Sciences | F12-GM117582 | Amicia D Elliott |
| National Institute of Mental Health | ZIA-MH002800 | Benjamin H White |
| NIDDK | | Carson C Chow |
| NIBIB | | Hari Shroff |

The funders had no role in study design, data collection and interpretation, or the decision to submit the work for publication.

### Author contributions

Amicia D Elliott, Conceptualization, Software, Formal analysis, Supervision, Funding acquisition, Investigation, Visualization, Methodology, Writing - original draft, Writing - review and editing; Adama Berndt, Software, Formal analysis, Investigation, Writing - review and editing; Matthew Houpert, Formal analysis, Investigation, Writing - review and editing; Snehashis Roy, Software, Formal analysis, Writing - review and editing; Robert L Scott, Resources, Writing - review and editing; Carson C Chow, Conceptualization, Supervision, Funding acquisition, Writing - review and editing; Hari Shroff, Conceptualization, Supervision, Funding acquisition, Visualization, Writing - review and editing; Benjamin H White, Conceptualization, Resources, Supervision, Funding acquisition, Investigation, Visualization, Methodology, Writing - original draft, Project administration, Writing - review and editing

### Author ORCIDs

Amicia D Elliott 
Carson C Chow 
Benjamin H White 

### Decision letter and Author response

Decision letter https://doi.org/10.7554/eLife.68656.sa1
Author response https://doi.org/10.7554/eLife.68656.sa2

## Additional files

### Supplementary files

- Supplementary file 1. Pupal neuromuscular anatomy.
- Supplementary file 2. Variability of phase parameters.
- Transparent reporting form

### Data availability

The source data for the figures and tables in this study are available at figshare (https://figshare.com/collections/Pupal_behavior_emerges_from_unstructured_muscle_activity_in_response_to_neuromodulation_in_*Drosophila*/5489637) and computer code is posted to https://github.com/BenjaminHWhite/muscle_activity (copy archived at https://archive.softwareheritage.org/swh:1:rev:66456f6ff61e8faa9fe4b442b91ef3fce3b178f9).

The following dataset was generated:

| Author(s) | Year | Dataset title | Dataset URL | Database and Identifier |
|---|---|---|---|---|
| Elliott AD, Berndt A, Houpert M, Roy S, Scott RL, Chow CC, Shroff H, White BH | 2021 | Pupal behavior emerges from unstructured muscle activity in response to neuromodulation in Drosophila | https://doi.org/10.6084/m9.figshare.c.5489637 | figshare, 10.6084/m9.figshare.c.5489637 |

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
