## [Decision Letter]

**Acceptance summary:**

The insect ecdysis sequence is a long-standing model for understanding the role of neuromodulators in organizing complex goal-directed blocks of behavior; here it is elegantly explored in *Drosophila*. Pupal ecdysis has benefits in that it occurs in the confines of the puparium and muscle activity of the entire animal can be followed through the entire behavioral sequence. The authors have developed tools to monitor both the responses of individual muscle fibers and the activity in the endplates of the motoneurons that innervate them during the course of the behavioral sequence. They have identified a discrete set of motor "syllables" that underlie the various movements and then begin to dissect how removal of different components of the peptide cascade results in the recruitment of these syllables and, hence, the behaviors.

**Decision letter after peer review:**

Thank you for submitting your article "Pupal behavior emerges from unstructured muscle activity in response to neuromodulation in *Drosophila*" for consideration by *eLife*. Your article has been reviewed by 4 peer reviewers, including Chris Q Doe as the Reviewing Editor and Reviewer #1 and the evaluation has been overseen by Ronald Calabrese as the Senior Editor. The following individuals involved in review of your submission have agreed to reveal their identity: James W Truman (Reviewer #2); Paul Taghert (Reviewer #3); Aref Zarin (Reviewer #4).

Essential Revisions (for the authors):

1) In figures 4, S4, and 5 you show GCaMP activity patterns, but it does not appear to be ratiometric with a muscle marker. How can you be sure the changes in GCaMP you are seeing are not due to muscle movement? I understand that the pupa is not translocating, but every muscle contraction will generate some movement in surrounding muscles. It is unrealistic to redo the experiments for ratiometric imaging, but you should explain how you minimize artifactual muscle activity (or loss of activity) due to local body wall movements.

2) The background for the paper could be improved for a general journal such as *eLife*. Pupal ecdysis in *Drosophila* is quite specialized and it would help to place it in context of other insect ecdyses. The complexity and temporal ordering of the "ecdysis sequence" have been nowhere better described than in the paper by Joe Carlson on cricket ecdysis (J.R.Carlson, (1977) J. Comp. Physiol. 115:299): it illustrates the three major phases (a preparatory, an ecdysis and an expansional phase [which Carlson breaks into two]) and how each phase involves the recruitment of diverse motor elements in a characteristic order but with scope for variability. These three phases occur in all ecdysis sequences that have been studied and correspond to the P1, P2 and P3 phases of pupal "ecdysis" in higher flies.

3) I found it very difficult to relate their findings back to the behaviors that they are trying to explain. The paper does a good job of describing the "trees" but it does not then pull back to place the trees in the context of the "forest". I found it very exciting that they could deconstruct the complex movements shown through the three major phases of the pupal ecdysis sequence into a small number of elemental components (their PMEs and individual muscle contractions that constitute their "syllables"). However, it was not obvious as to how these syllables were then combined to generate the 8 basic behaviors they illustrate in Figure 2A. Table S2 is critical because it summarizes the syllables that are involved in building these 8 basic movements. (Indeed, this table is so key that they may want to move it to the main text, rather than supplemental information!). What I find lacking, though, is clear description the phase relationships of the various syllables during each of these basic behaviors. An attempt to reassemble the behaviors is presented in Figure 7 for two of the behaviors. We are shown the order of appearance of the PMEs, but not their phase (and overlap?) relative to each other during a typical bout. As described below for the figures, I do not think that their color coding works very well, even if the images were large enough to resolve the detail. The authors might consider using a more traditional representation, such as seen in papers defining phases of muscle contraction during a locomotor cycles. This would be a phase diagram using blocks to show the phase and duration of each muscle contraction [in this case the authors would have syllables {individual muscles or PMEs}] during a cycle [variance could also be indicated at the ends of each block]. Although there may be unique events at the two ends of the body [which cannot be resolved in the figures], the authors still focus on a segmental unit in their representations. They could anchor their descriptions on a typical mid-body segment (e.g., A4) and show the timing of syllable recruitment during a behavioral bout.

*Reviewer #3 (Recommendations for the authors):*

The technical values of this work are very high and the argument that stereotyped sequences of muscle activation arise from smaller stochastically-produced elements (termed syllables) is novel and most interesting. Likewise, the manuscript effectively presents interesting observations on release activity of neuromodulatory inputs and relates these to long-standing ideas about checkpoint controls in ecdysis.

*Reviewer #4 (Recommendations for the authors):*

1) In the "Automated movement analysis" section in Methods, the last sentence in the first paragraph is missing a "was" ("The CNN model was then trained by …").

2) The keyword "neural circuits" might not be a good match to the purpose of this paper. The study examined thoroughly on muscle activity and the role of neuromodulation; however, it did not investigate multiple layers of neural circuits.

3) The authors could, in the introduction, briefly describe the role of ETHR and CCAP in *Drosophila*, especially them having only minor role in larval ecdysis but critical role in pupal ecdysis.

4) In Figure S4A, plotting both ΔF/F and ΔL/L over time might result in a better visual effect.

---

## [Author Response]

Essential Revisions (for the authors):1) In figures 4, S4, and 5 you show GCaMP activity patterns, but it does not appear to be ratiometric with a muscle marker. How can you be sure the changes in GCaMP you are seeing are not due to muscle movement? I understand that the pupa is not translocating, but every muscle contraction will generate some movement in surrounding muscles. It is unrealistic to redo the experiments for ratiometric imaging, but you should explain how you minimize artifactual muscle activity (or loss of activity) due to local body wall movements.

We fully appreciate the reviewers’ concern and acknowledge the merits of ratiometric imaging in correlating Ca^++^ signals with muscle movements. Although technical constraints prevented us from routinely collecting simultaneous images in two channels, we took several steps to insure that the Ca^++^ signals we measured were correctly associated with the muscles in which they were generated and that changes in signal were not artifacts of muscle displacement due to generalized movements of the body wall. First, because the pupa has half the muscles of the larva, it was possible to measure the GCaMP signals in ROIs that included only the muscle of interest, and not neighboring muscles. Such ROIs were manually identified and the signal quantified for all of the single muscle data reported in the figures referred to. We are thus confident that there is no cross-contamination of signal from adjacent muscles during movement. Second, artifacts due to shifts in image plane during movement were avoided by using low magnification (4x) and widefield illumination. This insured that light was collected not from a single image plane, but from a large volume that included the side of the body wall facing the objective and all of its muscles. Ca^++^ signals from muscles in the body wall on the opposite side of the animal were excluded from the volume due to scattering and occlusion by internal tissues, which we confirmed by imaging z stacks. We have updated the Methods to clarify our methodology (ll. 980-989 [1056-1065]).

2) The background for the paper could be improved for a general journal such as eLife. Pupal ecdysis in *Drosophila* is quite specialized and it would help to place it in context of other insect ecdyses.

We thank the reviewer for this suggestion and have expanded the background on ecdysis research for a wider audience. We have expanded the Introduction (ll. 51-89 [51-93]) to describe the broader research on insect ecdysis, including the paradigmatic and detailed descriptions provided by Carlson for crickets and Hughes for locusts. Incorporating the suggestion of Reviewer 3 below, we also take the opportunity to place the work on ecdysis in the context of behavioral endocrinology. We note the long history of work on the hormonal control of behavior in both invertebrate and vertebrate preparations of which ecdysis research is a part.

3) I found it very difficult to relate their findings back to the behaviors that they are trying to explain. The paper does a good job of describing the "trees" but it does not then pull back to place the trees in the context of the "forest". I found it very exciting that they could deconstruct the complex movements shown through the three major phases of the pupal ecdysis sequence into a small number of elemental components (their PMEs and individual muscle contractions that constitute their "syllables"). However, it was not obvious as to how these syllables were then combined to generate the 8 basic behaviors they illustrate in Figure 2A. Table S2 is critical because it summarizes the syllables that are involved in building these 8 basic movements. (Indeed, this table is so key that they may want to move it to the main text, rather than supplemental information!). What I find lacking, though, is clear description the phase relationships of the various syllables during each of these basic behaviors. An attempt to reassemble the behaviors is presented in Figure 7 for two of the behaviors. We are shown the order of appearance of the PMEs, but not their phase (and overlap?) relative to each other during a typical bout. As described below for the figures, I do not think that their color coding works very well, even if the images were large enough to resolve the detail. The authors might consider using a more traditional representation, such as seen in papers defining phases of muscle contraction during a locomotor cycles. This would be a phase diagram using blocks to show the phase and duration of each muscle contraction [in this case the authors would have syllables {individual muscles or PMEs}] during a cycle [variance could also be indicated at the ends of each block]. Although there may be unique events at the two ends of the body [which cannot be resolved in the figures], the authors still focus on a segmental unit in their representations. They could anchor their descriptions on a typical mid-body segment (e.g., A4) and show the timing of syllable recruitment during a behavioral bout.

We appreciate the Reviewer’s detailed comments on this important point and are grateful for the suggestion that we represent the phases of syllable activity during movement generation. Based on this and other reviewers’ comments, we recognize that our attempt to bridge the gap between syllables and movements in the summary figure (i.e. Figure 7) of our original manuscript was unsatisfying. In the revised manuscript, we present results from the kind of phase analysis suggested by the reviewer and think that these results both strengthen the link between the syllables (“trees”) and the movements (“forest”) and illustrate this link in a relatively straight-forward way.

As the reviewer alludes to in his comments, analyzing syllable phases is subject to several constraints. In the absence of an automated method for syllable detection, the required measurements of onset and offset times for each syllable need to be made manually—a daunting task for all syllables in the entire pupal ecdysis sequence! In addition, many pupal movements, particularly those of P3, involve distinct activity in different segments—or even in different halves of the same segment. There is thus no single, “representative” hemisegment on which to conduct the analysis. This is quite different from the peristaltic contractions of larval locomotion, which can be conveniently captured by analyzing muscle contractions in a single hemisegment. To accommodate these constraints, we performed a restricted analysis that focused on the relatively simple movements of P1 and P2. These movements involve muscle activity that is similar across hemisegments (as for the RollCon and Swing) or similar within the segments of the posterior compartment (as for the Lift). By confining the analysis to two hemisegments, one (A6) in the posterior compartment and one (A4) in the anterior compartment, we could thus resolve syllable dynamics for both movements of P1 and could monitor the progression of the P2 Swing as it traverses both the posterior and anterior compartments. To do so, we measured the phasic activity of all syllables visible from the lateral view for A6 and A4 in a set of representative bouts of P1 and P2. These analyses are now presented as phase diagrams in Figure 7D and E of the revised manuscript. As suggested by the Reviewer, we used blocks to show the phase and duration of each syllable during a bout and indicated the standard deviations at the ends of each block. In the accompanying text (ll. 246-274 [273-301]), we describe how the phasic activity of the syllables gives rise to the movements executed. We also now provide a supplemental figure (Figure 7—figure supplement 1) that illustrates the syllable dynamics of a Swing movement for muscles of all hemisegments of the pupa. In addition, we have taken the reviewer’s suggestion to elevate the former Table S2, describing the syllable composition of all movements, to Table 1 in the main text. We have also tried to improve the color-coding in figures throughout the manuscript to resolve potential ambiguities. In an effort to rationalize the use of color, we chose from the palette of colors recommended for the color-blind.

Reviewer #4 (Recommendations for the authors):(1) In the "Automated movement analysis" section in Methods, the last sentence in the first paragraph is missing a "was" ("The CNN model was then trained by …").

Thanks very much for catching this detail. We have inserted the missing word (l. 1078 [1154]).

(2) The keyword "neural circuits" might not be a good match to the purpose of this paper. The study examined thoroughly on muscle activity and the role of neuromodulation; however, it did not investigate multiple layers of neural circuits.

We appreciate the reviewer’s point, but certainly hope that our paper will be of interest to others who are mapping neural circuits. Motor neurons represent the final common path for all circuit activity and our goal was to define activity in this path as an entry point for understanding activity at higher levels in the pupal ecdysis circuit. We hope that others will see the merits of using muscle activity as a proxy for motor neuron activity to gain traction on how circuit-level activity generates behavior.

(3) The authors could, in the introduction, briefly describe the role of ETHR and CCAP in *Drosophila*, especially them having only minor role in larval ecdysis but critical role in pupal ecdysis.

In the expanded description of ecdysis requested by several reviewers, we now outline the roles in ecdysis played by neurons targeted by ETH and CCAP in the Introduction (ll. 79-86 [80-90]).

(4) In Figure S4A, plotting both ΔF/F and ΔL/L over time might result in a better visual effect.

To show how these values change over time, we have now separated them out by phase (Figure 8—Figure supplement 1A in the revised draft).